# Efficient Truncated Linear Regression with Unknown Noise Variance

**Constantinos Daskalakis**
*EECS and CSAIL, MIT
costis@csail.mit.edu

**Patroklos Stefanou**
EECS and CSAIL, MIT
stefanou@mit.edu

**Rui Yao**
EECS and CSAIL, MIT
rayyao@mit.edu

**Manolis Zampetakis** [†]
EECS, UC Berkeley
mzampet@berkeley.edu

## Abstract

Truncated linear regression is a classical challenge in statistics, wherein a label, $y = w^T x + \varepsilon$, and its corresponding feature vector, $x \in \mathbb{R}^k$, are only observed if the label falls in some subset $S \subseteq \mathbb{R}$; otherwise the existence of the pair $(x, y)$ is hidden from observation. Linear regression with truncated observations has remained a challenge, in its general form, since the early works of Tobin (1958); Amemiya (1973). When the distribution of the error is normal with known variance, recent work of Daskalakis et al. (2019) provides computationally and statistically efficient estimators of the linear model, $w$. In this paper, we provide the first computationally and statistically efficient estimators for truncated linear regression when the noise variance is unknown, estimating both the linear model and the variance of the noise. Our estimator is based on an efficient implementation of Projected Stochastic Gradient Descent on the negative log-likelihood of the truncated sample. Importantly, we show that the error of our estimates is asymptotically normal, and we use this to provide explicit confidence regions for our estimates.

## 1 Introduction

A common challenge facing statistical estimation is the systematic omission of relevant data from the data used to train models. Data omission is a prominent form of dataset bias, which may occur for a variety of reasons, including incorrect experimental design or data collection campaigns, which prevent certain sub-populations from being observed, instrument errors or saturation, which make certain measurements unreliable, societal biases, which may suppress the realization of certain samples, as well as legal or privacy constraints, which might prevent the use of some data. In turn, it is well-understood that failure to account for the systematic omission of data in statistical inference may lead to incorrect models, and this has motivated a long line of research in statistics, econometrics, and a range of other theoretical and applied fields, targeting statistical inference that is robust to missing data; see e.g. Galton (1897); Pearson (1902); Pearson & Lee (1908); Fisher (1931); Tukey (1949); Tobin (1958); Amemiya (1973); Hausman & Wise (1977); Heckman (1979); Breen et al. (1996); Hajivassiliou & McFadden (1998); Mohan et al. (2013); Balakrishnan & Cramer (2014).

In this paper, we revisit the classical challenge of *truncated linear regression*. As in the standard linear regression setting, we assume that the world produces pairs $(x^{(i)}, y^{(i)})_i$, where each $x^{(i)} \in \mathbb{R}^k$

---

[*]CD and PS were supported by NSF Awards IIS-1741137, CCF-1617730, and CCF-1901292, by a Simons Investigator Award, by the Simons Collaboration on the Theory of Algorithmic Fairness, by a DSTA grant, 11 and by the DOE PhILMs project (No. DE-AC05-76RL01830)

[†]MZ was supported by NSF DMS-2023505 (FODSI)

is a feature vector drawn from some distribution $\mathcal{P}$ and its corresponding label $y^{(i)}$ is linearly related to $x^{(i)}$ according to $y^{(i)} = w^{*T}x^{(i)} + \varepsilon^{(i)}$, for both some unknown coefficient vector $w^* \in \mathbb{R}^k$ and random noise $\varepsilon^{(i)} \sim \mathcal{N}(0, \sigma^{*2})$, where $\sigma^{*2}$ is unknown and $\varepsilon^{(i)}$ is $i.i.d$ over all $i$. The difference to the standard setting is that we do not get to observe all pairs $(x^{(i)}, y^{(i)})$ that the world produces. Rather, a pair $(x^{(i)}, y^{(i)})$ is only observed if $y^{(i)} \in S$, where $S \subset \mathbb{R}$ is a fixed "observation set," that is known or given to us via oracle access, i.e. we can query an oracle about whether or not some point belongs to $S$. If $y^{(i)} \notin S$, then $(x^{(i)}, y^{(i)})$ is removed from observation. Our goal is to estimate $w^*$ and $\sigma^{*2}$ from the pairs that survived removal, called "truncated samples."

**Overview of our results.**   While both truncated and censored linear regression are fundamental problems, which commonly arise in practice and have been studied in the literature since at least the 1950s (Tobin, 1958; Amemiya, 1973), there are no known statistical inference methods that are computationally efficient and enjoy statistical rates that scale with the dimension $k$ of the problem. Recent work of Daskalakis et al. (2019) has obtained such methods in the simpler setting where the noise variance $\sigma^{*2}$ is known. Our goal here is to extend those results in two important ways. First, we study settings where $\sigma^{*2}$ is unknown, providing computationally and statistically efficient estimators for the joint estimation of $w^*$ and $\sigma^{*2}$. Moreover, we establish the asymptotic normality of our estimators, and use this to derive confidence regions for our estimators. To achieve these stronger results we make some additional assumptions compared to Daskalakis et al. (2019). Namely, we assume that the covariates are sampled from some prior distributions, and we also make a stronger assumption about the survival probability of our model as we discuss below. Other works that consider learning and regression problems with censored or truncated samples are Plevrakis (2021); Fotakis et al. (2020); Moitra et al. (2021); Ilyas et al. (2020).

We next provide informal statements of our results, postponing formal statements to Sections 3 and 5. It is important to note that, as shown in prior work (Daskalakis et al., 2018, 2019), for the model parameters to be identifiable from truncated samples, we need to place some assumptions that limit how aggressively the data is truncated. The assumptions we make are listed below, and stated more precisely in Section 2. They combine normalization assumptions (Assumption 1.1), assumptions that are needed even without truncation (Assumption 1.3), and assumptions that are needed for identifiability purposes in the presence of truncation (Assumption 1.2).

**Assumption 1.1.** We assume that $\|w^*\|_2^2 \leq \beta$ and $\|x^{(i)}\|_2^2 \leq 1$ for all $i$.

**Assumption 1.2.** For all $(x^{(i)}, y^{(i)})$ in the truncated dataset, conditioning on $x^{(i)}$, the probability that $w^{*T}x^{(i)} + \varepsilon \in S$, with respect to the randomness in $\varepsilon \sim \mathcal{N}(0, \sigma^{*2})$, is larger than some $a$.

**Assumption 1.3.** If $(x^{(i)}, y^{(i)})_{i=1}^n$ is the truncated dataset, we assume that the feature covariance matrix, $X = \frac{1}{n}\sum_{i=1}^n x^{(i)}x^{(i)^T}$, satisfies: $\mathbb{E}_{x \sim \mathcal{P}}[X] = \mathbb{E}_{x \sim \mathcal{P}}[xx^T] \succeq b \cdot I$ for some $b$.

With these assumptions we prove our main results for the joint estimation of $w^*$ and $\sigma^{*2}$, listed as Informal Theorems 1.4–1.5 below, and respectively Theorems 3.2 and 5.1 in later sections.

**Informal Theorem 1.4** (Estimation). *Suppose we are given $n$ samples $(x^{(i)}, y^{(i)})_{i=1}^n$ from the truncated linear regression model with parameters $w^*$, $\sigma^{*2}$, and suppose Assumptions 1.1–1.3 hold. Then running Projected Stochastic Gradient Descent (Algorithm 1) on the conditional negative log-likelihood of the sample, using the projection set described in Definition 3.1, we obtain with probability at least $2/3$ (which can be boosted to as large a constant as desirable) estimations $\hat{w}, \hat{\sigma}^2$ such that $\|\hat{w} - w^*\|_2 + |\hat{\sigma}^2 - \sigma^{*2}| \leq \varepsilon$, as long as $n \geq \frac{C_1 \log^4(1/\varepsilon)}{\varepsilon^4}$ or $n \geq \frac{C_2}{\varepsilon^2}$ (depending on the learning rate), where $C_1$ depends polynomially on $\beta$, $1/a$ and $1/b$, and $C_2$ depends linearly in $1/b$ and exponentially in $\beta/a$, where $\beta$, $a$ and $b$ are respectively the constants in Assumptions 1.1, 1.2 and 1.3.*

**Informal Theorem 1.5** (Asymptotic Normality). *Let $\hat{w}, \hat{\sigma}^2$ be the estimates of Informal Theorem 1.4. There exists a matrix $A$ that depends on $\hat{w}, \hat{\sigma}^2$ (and is explicitly provided in Theorem 5.1) such that the random vector $\sqrt{n}(\binom{\hat{w}}{\hat{\sigma}^2} - \binom{w^*}{\sigma^{*2}})$ converges asymptotically to a normal distribution with zero mean and covariance matrix $A$. Hence the $1 - \alpha$ confidence region of $(\hat{w}, \hat{\sigma}^2)$ can be computed using the matrix $A$ and the quantiles of the chi-squared distribution.*

**Techniques.**   Our estimates are computed by running Projected Stochastic Gradient Descent (PSGD) on the population conditional negative log-likelihood of the truncated sample; a convex function of

the parameters, after an appropriate reparameterization. We provide two analyses of PSGD resulting in two different bounds that are more effective in a different range of parameters, as we discuss next.

Our first analysis hinges on establishing a lower bound on the convexity of the conditional negative log-likelihood function. We can't however, expect that a strong bound holds over the whole parameter domain. For this reason, we add a projection step to our method, identifying a projection set wherein we show that the true parameters lie and wherein the conditional negative log-likelihood is strongly convex. This approach is similar to that of Daskalakis et al. (2019), except that the unknown noise variance introduces many challenges in identifying an appropriate projection set. The main result bounding the strong convexity of the conditional negative log likelihood function within the projection set is stated as Theorem 4.1. The main disadvantage of this analysis is that, when the variance is unknown, the bound on the strong convexity of the conditional negative log-likelihood becomes small and the resulting dependence of the sample size on the parameters $\beta$ and $1/a$ of the problem is exponential. On the other hand, the dependence of the sample size on $1/b$ is linear and its dependence on the required estimation accuracy is $\Theta(1/\varepsilon^2)$, which is optimal. Our other analysis follows an approach that has not been pursued in prior work, and in particular does not rely on a global lower bound on the strong convexity of the conditional negative log-likelihood. Instead, we only bound the strong convexity at the optimum and use an upper bound on the smoothness of the objective function to prove the convergence of PSGD in terms of function value. Again, it is impossible to prove a bound on the smoothness over the whole domain, because large noise variance can lead to very bad smoothness bounds. Thus we use the same projection set as above and show that, within this set, we can get an upper bound on the smoothness that depends polynomially on all parameters, $\beta, 1/a, 1/b$, of the problem. The main drawback of this method is that the dependence on the accuracy is $\Theta(\log^4(1/\varepsilon)/\varepsilon^4)$. It is worth noting that obtaining an algorithm whose sample complexity has an optimal $\Theta(1/\varepsilon^2)$ dependence on the estimation accuracy, and also scales polynomially in $1/a$ is a fundamental bottleneck, which is already present in the much simpler special case of our problem of estimating the variance of a 1-dimensional truncated Gaussian, studied in Daskalakis et al. (2018).[3]

Finally, to help build intuition around the intricacies in estimating truncated linear regression models with unknown noise variance, let's consider why natural approaches of reducing this problem to estimating truncated linear regression models with *known* noise variance – which can be done using the algorithm of Daskalakis et al. (2019) – fail. A simple such reduction might compute the empirical variance $\sigma_0^2$ of the truncated sample, and plug that into the algorithm of Daskalakis et al. (2019) to estimate the linear model, pretending that $\sigma_0^2$ is the true noise variance. A more sophisticated approach might grid over candidate noise variances, plug each of these candidates into the algorithm of Daskalakis et al. (2019) to estimate a linear model, and then select the (linear model, noise variance) pair that is most consistent, e.g. the pair whose noise variance is closest to the empirical variance of the residuals between the linear model's predictions and the observed labels. Both of these approaches fail, as suggested by the example below. Suppose that the ground truth is $y = w^*x + w_0^* + \varepsilon$, where $x$ is one-dimensional, $(w^*, w_0^*) = (0, 0)$, and the noise variance is $\sigma^{*2} = 1$. Suppose also that we create a truncated dataset by, repeatedly, sampling $x_i$ from $\mathcal{N}(0, 1)$, generating $y_i$ according to the ground truth, and truncating the resulting pair $(x_i, y_i)$ unless $y_i \geq \tau$, where $\tau$ is some threshold. It is clear that, if we guess the true noise variance to be the empirical one, then as $\tau$ gets bigger, the empirical variance gets smaller, thus the linear model that would be estimated by treating the empirical variance as the true noise variance would have intercept that is close to $\tau$ rather than $0$. So this naive approach would fail. Let us consider the more sophisticated one that grids over candidate noise variances. In particular, let us compare what happens when (1) we guess the noise variance to be $\sigma_1^2 = \sigma^{*2}$, and run the algorithm of Daskalakis et al. (2019) with this guess to get an estimate $(\hat{w}, \hat{w}_0)$; and (2) when we guess the noise variance to be $\sigma_2^2 \equiv \sigma_0^2$ (the empirical variance) and run the algorithm of Daskalakis et al. (2019) with this guess to get an estimate $(\hat{w}', \hat{w}_0')$. It is easy to see that, as $\tau$ gets large, $\left| \sigma^{*2} - \frac{1}{N}\sum_i (y_i - \hat{w}x_i - \hat{w}_0)^2 \right| \gg \left| \sigma_0^2 - \frac{1}{N}\sum_i (y_i - \hat{w}'x_i - \hat{w}_0')^2 \right|$ (here, $N$ is the number of samples. This holds even if $N$ goes to infinity). Thus the selection criterion proposed above will fail to prefer (1) over (2), even though the guess of the noise variance under (1) is the correct one. Our approach in this paper amounts to using a different criterion, namely selecting the pair with the largest conditional log-likelihood. But analyzing when and how well this works, as well as obtaining an efficient algorithm to find the best pair is the main contribution of this work.

---

[3]While Daskalakis et al. (2018) does not explicitly analyze the dependence of the sample complexity on $\alpha$; by tracing through their bounds, one can see that the dependence of the sample complexity on $\alpha$ is exponential.

**Roadmap.** We present the truncated linear regression setting and our assumptions in Section 2. We state and prove our main estimation results Section 3 and 4, and prove the asymptotic normality of our estimators in Section 5. Finally, in Section 6 we assess the performance of our methods.

## 2 Models and assumptions

We study the truncated linear regression setting studied in Daskalakis et al. (2019), with the additional complexity that the variance of the noise model is unknown. In particular, let $S \subset \mathbb{R}$ be a measurable subset of the real line, to which oracle access is provided. Assume we have $N = 3n$ truncated samples $(x^{(i)}, y^{(i)})_{i=1}^N$ (we will split the samples into three, see 4.5), each generated according to the following procedure, for some unknown $w^* \in \mathbb{R}^k$ and $\sigma^* \in \mathbb{R}$ that are fixed across different $i$'s:

1. sample $x^{(i)}$ from some distribution $\mathcal{P}_0$ with support $\mathrm{supp}(\mathcal{P}_0) \subseteq \mathbb{R}^k$;

2. generate $y^{(i)}$ according to the following model, where $\varepsilon \sim \mathcal{N}(0, \sigma^{*2})$;

$$y^{(i)} = w^{*T} x^{(i)} + \varepsilon \tag{2.1}$$

3. if $y^{(i)} \in S$ then return $(x^{(i)}, y^{(i)})$ as the $i$th sample, otherwise repeat from step 1.

After truncation, denote the distribution of $x$ to be $\mathcal{P}$ with the $\mathrm{supp}(\mathcal{P}) = \mathrm{supp}(\mathcal{P}_0)$. The sampled $x$ is still $i.i.d$, so we could only consider the distribution $\mathcal{P}$. As already established in prior work (Daskalakis et al., 2018, 2019; Kontonis et al., 2019), some assumptions must be placed on $S$ and its interaction with the unknown parameters $(w^*, \sigma^{*2})$ and the $x^{(i)}$'s in order for the parameters to be identifiable. We state our assumptions after a useful definition.

**Definition 2.1** (Survival Probability). Let $S \subseteq \mathbb{R}$ be measurable, $x, w \in \mathbb{R}^k$, and $\sigma \in \mathbb{R}$. We define the survival probability $\alpha(w, \sigma, x, S)$ as $\alpha(w, \sigma, x, S) = \mathbb{P}\{Y \in S\}$; where $Y \sim \mathcal{N}(w^T x, \sigma^2)$. When $S$ is clear from context we may omit $S$ from the arguments of $\alpha$ and simply write $\alpha(w, \sigma, x)$.

We are now ready to state our assumptions. Our first assumption gives an upper bound on the norm of the true parameters and the feature vectors.

**Assumption 2.2** (Normalization/Data Generation). Assume that $\|w^*\|_2^2 \leq \beta$, and $\|x\|_2^2 \leq 1$ for all $x \in \mathrm{supp}(\mathcal{P})$. We also assume oracle access for set $S$, i.e. we have an oracle which determines, for any $y$, whether $y \in S$.

Our next assumption is the survival probability lower bound following Daskalakis et al. (2018, 2019).

**Assumption 2.3** (Constant Survival Probability). For all $x \in \mathrm{supp}(\mathcal{P})$, we have $\alpha(w^*, \sigma^*, x) > a$.

Our final assumption is about the spectrum of the covariance matrix of the features and is classical in some other linear regression settings, such as in Daskalakis et al. (2019).

**Assumption 2.4** (Thickness of Feature Covariance). Let $X = \frac{1}{n} \sum_{i=1}^n x^{(i)} x^{(i)T}$ be the $k \times k$ feature vector covariance matrix. We assume that, for some positive real number $b$, we have $\mathbb{E}_{x \sim \mathcal{P}}[X] = \mathbb{E}_{x \sim \mathcal{P}}[xx^T] \succeq b \cdot I$. Here we abuse the notation that the $(x^{(i)}, y^{(i)})_{i=1}^n$ are samples used for PSGD.

## 3 Main Result

When the variance parameter $\sigma^{*2}$ is unknown, the conditional log-likelihood function of the plain linear regression becomes non-concave, and is therefore not straightforward to optimize. For this reason, we follow the common practice of reparameterizing the problem with respect to the parameters $v = w/\sigma^2$ and $\lambda = 1/\sigma^2$. The algorithm that we use is Projected SGD on the population negative log-likelihood function $\bar{\ell}(v, \lambda)$ as presented in Algorithm 1. We formally define the loss function $\bar{\ell}$ in Section 4.1.

Our goal is to apply the above algorithm to the conditional negative log-likelihood function of the truncated linear regression model. To execute the above algorithm in polynomial time, we need to solve the following algorithmic problems.

(a) **initial feasible point:** compute an initial feasible point in some projection set $D$,

---

**Algorithm 1** Projected SGD on $\bar{\ell}$

---

**Input:** $(x^{(i)}, y^{(i)})$ for $i \in [n]$, learning rates $\eta_t$

1: $(w_0, \sigma_0^2) \leftarrow$ OLS estimates using other $n$ samples; $(q, \lambda^{(0)}) \leftarrow (w_0/\sigma_0^2, 1/\sigma_0^2)$
                     $\triangleright$ *(a) Initialization part 1*

2: In terms of $\sigma_0^2$ define convex set $D$ over pairs $(v, \lambda)$ as described in Definition 3.1

3: $v^{(0)} \leftarrow \operatorname{argmin}_{v:(v,\lambda^{(0)})\in D} \|v - q\|_2$        $\triangleright$ *(a) Initialization part 2*

4: **for** $t = 1, \ldots, n$ **do**

5:   Sample gradient $u^{(t)}$ such that $\mathbb{E}\left[u^{(t)}|(v, \lambda)^{(t-1)}\right] = \nabla\bar{\ell}(v^{(t-1)}, \lambda^{(t-1)})$
          $\triangleright$ *(b) Unbiased Gradient Estimate, sampling method see Section B.1*

6:   $(v^{(t)}, \lambda^{(t)}) \leftarrow (v^{(t-1)}, \lambda^{(t-1)}) - \eta_t \cdot u^{(t)}$

7:   $(v^{(t)}, \lambda^{(t)}) \leftarrow \arg\min_{(v,\lambda)\in D}\left\|(v, \lambda) - (v^{(t)}, \lambda^{(t)})\right\|_2$    $\triangleright$ *(c) Efficient Projection*

8: **end for**

9: $(\hat{v}, \hat{\lambda}) \leftarrow (v^{(n)}, \lambda^{(n)})$; **return** $(\hat{w}, \hat{\sigma}^2) \leftarrow \left(\hat{v}/\hat{\lambda}, 1/\hat{\lambda}\right)$

---

 (b) **unbiased gradient estimation:** sample an unbiased estimate of $\nabla\bar{\ell}(v^{(t-1)}, \lambda^{(t-1)})$,

 (c) **efficient projection:** design an algorithm to project to the set $D$.

Solving the computational problems (a)-(c) is the first step in the proof of our main result. First we define the appropriate projection set to use in our algorithm.

**Definition 3.1** (Projection Set). We define the projection set to be the $(v, \lambda)$ satisfying

$$D = \left\{(v, \lambda) = \left(\frac{w}{\sigma^2}, \frac{1}{\sigma^2}\right) \in \mathbb{R}^k \times \mathbb{R} \;\Big|\; \frac{1}{8(5 + 2\log(1/a))} \leq \frac{\sigma^2}{\sigma_0^2} \leq \frac{96}{a^2}, \|w\|_2^2 \leq \beta\right\}$$

where $\sigma_0^2$ is the estimated variance from solving the classical linear regression problem (using $n$ samples) on the data ignoring truncation.

This projection set contains two parts: the first part that restricts the weights and the second part that restricts the variance. For the second part, we use the variance predicted using ordinary least squares.

Now we are ready to formally state our main theorem about the estimation of the parameters $w$ and $\sigma$ from truncated samples under the Assumptions 2.2- 2.4.

**Theorem 3.2.** *Let* $(x^{(1)}, y^{(1)}), \cdots, (x^{(n)}, y^{(n)})$ *be* $n$ *samples from the linear regression model* (2.1) *with ground truth* $w^*, \sigma^{*2}$. *If Assumptions* 2.2- 2.4 *hold, then we can instantiate the learning rates used in Algorithm 1 such that, with success probability* $\geq 2/3$, *the output estimates* $\hat{w}, \hat{\sigma}^2$ *satisfy either one of the following bounds:*

$$\|\hat{w} - w^*\|_2 + \left|\hat{\sigma}^2 - \sigma^{*2}\right| \leq \frac{\operatorname{poly}(\frac{\sigma^* \cdot \beta}{a \cdot b}, \frac{1}{a \cdot \sigma^*}) \cdot \log(n)}{n^{1/4}} \tag{3.1}$$

$$\|\hat{w} - w^*\|_2 + \left|\hat{\sigma}^2 - \sigma^{*2}\right| \leq \frac{\operatorname{poly}(\sigma^*, \frac{1}{b}) \cdot \exp\left(\operatorname{poly}(\frac{\beta}{a \cdot \sigma^*})\right)}{\sqrt{n}} \tag{3.2}$$

*for some polynomials that we make explicit in the detailed proof of the theorem (Section B.7).*

*Remark* 3.3. It is easy to see that the probability of success in Theorem 3.2 can be boosted to $1 - \delta$ with an additional cost of order $\log(1/\delta)$ in the rates. We can achieve this using a folklore boosting technique for parameter estimation problems. In particular, we can run the algorithm of Theorem 3.2 $\log(1/\delta)$ times independently and then we can pick the estimate that is $\varepsilon$ close to at least the half of the rest of the estimates, where $\varepsilon$ is the target estimation error. It is easy to see that with this boosting technique we can get error at most $2\varepsilon$ with probability of success at least $1 - \delta$.

To prove Theorem 3.2 we provide two different analyses of the Algorithm 1. The first analysis uses only the fact that the stochastic gradients of the population conditional negative log-likelihood function have bounded second moment and is based on the following theorem.

**Theorem 3.4** (Theorem 2 of Shamir & Zhang (2013)). *Let* $\bar{\ell}$ *be a convex function with a minimizer* $(v^*, \lambda^*)$ *and suppose that there exists a constant* $\rho$ *such that*

(i) **bounded step variance :** $\mathbb{E}\left[\left\|u^{(t)}\right\|_2^2\right] \leq \rho^2$ *where $u^{(t)}$ is the sampled step variance in algorithm 1,*

(ii) **bounded domain:** $\max_{(v,\lambda)\in D}\|(v-v^*,\lambda-\lambda^*)\|_2^2 \leq \rho^2$,

(iii) **feasibility of optimal:** *the minimizer $(v^*,\lambda^*)\in D$*

*then if we apply Algorithm 1 on $\bar{\ell}$ with learning rates $\eta_t = c/\sqrt{t}$ where $c$ is an absolute constant then for every $t \in [n]$ it holds that $\mathbb{E}\left[\bar{\ell}(v^{(t)},\lambda^{(t)}) - \bar{\ell}(v^*,\lambda^*)\right] \leq \frac{\text{poly}(\rho)\cdot\log(t)}{\sqrt{t}}$.*

For the second analysis of Algorithm 1 we also need to bound the strong convexity of $\bar{\ell}$. This is the reason that we get the exponential dependence on some of the parameters of the problem, but we can also get a better consistency rate. For this second analysis we use the following theorem.

**Theorem 3.5** (Lemma 1 of Rakhlin et al. (2011)). *Let $\bar{\ell}$ be a convex function with a minimizer $(v^*,\lambda^*)$ and suppose that there exist constants $\rho$, $\zeta$ such that*

(i) **bounded step variance:** $\mathbb{E}\left[\left\|u^{(t)}\right\|_2^2\right] \leq \rho^2$ *where $u^{(t)}$ is the sampled step variance in algorithm 1*

(iii) **feasibility of optimal:** *the minimizer $(v^*,\lambda^*)\in D$*

(iv) **strong convexity:** $\bar{\ell}$ *is $\zeta$-strongly convex*

*then if we apply Algorithm 1 on $\bar{\ell}$ with learning rates $\eta_t = 1/(\zeta \cdot t)$, then for every $t \in [n]$ it holds that*

$$\mathbb{E}\left[\left\|(v^{(t)},\lambda^{(t)}) - (v^*,\lambda^*)\right\|_2^2\right] \leq \frac{4\rho^2}{\zeta^2 \cdot t}.$$

In sum, the main challenges for proving Theorem 3.2 are solving computational problems (a)–(c) and proving mathematical properties (i)–(iv).

## 4 Overview of the proof of Theorem 3.2.

We provide an overview of the proof of Theorem 3.2, postponing most technical details to the supplementary material. The main steps of the proof are the following.

1. In Section 4.1 we derive the negative population conditional log-likelihood, its gradient, and its Hessian matrix.
2. In Section 4.2 we discuss the computational problems (a) - (c).
3. In Section 4.3 we formally state our results that prove the properties (i), (ii), and (iv).
4. In Section 4.4 we state our results for property (iii), the feasibility of the optimal solution.
5. Finally in Section 4.5 we use all the above results to complete the proof of 3.2.

### 4.1 The Negative Population Conditional Log-Likelihood Function of Truncated Linear Regression

In this section we define the objective function that we use to apply our PSGD algorithm. This objective function is derived by taking the negative expected value of the log-likelihood of $y$ conditional on the value of $x$. Given a sample $(x,y)$, the negative conditional log-likelihood that $y$ is a sampled from the truncated linear regression with parameters $w, \sigma^2$ given the value of $x$ is equal to

$$\ell(w,\sigma^2;x,y) = \frac{1}{2\sigma^2}(y^2 - 2yw^Tx) + \log\left(\int_S \exp(-\frac{1}{2\sigma^2}(z^2 - 2zw^Tx))\mathrm{d}z\right)$$

If we reparameterize $\ell$ with $\lambda = 1/\sigma^2$ and $v = w/\sigma^2$, we have

$$\ell(v,\lambda;x,y) = \frac{1}{2}(\lambda y^2 - 2yv^Tx) + \log\left(\int_S \exp(-\frac{1}{2}(\lambda z^2 - 2zv^Tx))\mathrm{d}z\right) \qquad (4.1)$$

We define the distributions $F_x = \mathcal{N}(w^{*T}x, \sigma^{*2}, S)$ and $Q_x = \mathcal{N}(w^T x, \sigma^2, S)$, and the *negative population conditional log-likelihood function* $\bar{\ell}$ as follows

$$\bar{\ell}(v, \lambda) = \mathop{\mathbb{E}}_{x \sim \mathcal{P}}\left[\mathop{\mathbb{E}}_{y \sim F_x}[\ell(v, \lambda; x, y)]\right]. \tag{4.2}$$

We can then easily compute

$$\frac{\partial \bar{\ell}}{\partial v} = \mathop{\mathbb{E}}_{x \sim \mathcal{P}}\left[\mathop{\mathbb{E}}_{z \sim Q_x}[z \cdot x] - \mathop{\mathbb{E}}_{y \sim F_x}[y \cdot x]\right] \quad \frac{\partial \bar{\ell}}{\partial \lambda} = \frac{1}{2} \mathop{\mathbb{E}}_{x \sim \mathcal{P}}\left[\mathop{\mathbb{E}}_{y \sim F_x}[y^2] - \mathop{\mathbb{E}}_{z \sim Q_x}[z^2]\right],$$

From these we also compute the Hessian matrix which is

$$\mathbf{H}(v, \lambda) = \mathop{\mathbb{E}}_{x \sim \mathcal{P}}\left[\begin{pmatrix} \mathrm{Var}_{z \sim Q_x}[z]xx^T & -\mathrm{Cov}_{z \sim Q_x}[\frac{1}{2}z^2, z]x \\ -\mathrm{Cov}_{z \sim Q_x}[\frac{1}{2}z^2, z]x^T & \mathrm{Var}_{z \sim Q_x}[\frac{1}{2}z^2] \end{pmatrix}\right] \tag{4.3}$$

Here, $\mathbf{H}$ is the expectation of covariance matrices, which is positive definite. Thus, $\bar{\ell}$ is convex.

## 4.2 Computational Problems

The computational problems (a) - (c) can be tackled in the following way

(a) We start with computing $(w_0, \sigma_0)$ by applying OLS to the truncated data.

(b) As we see from the expression of the gradient of $\bar{\ell}$, we can compute an unbiased estimate of the gradient using one of our samples $(x^{(i)}, y^{(i)})$, our sample access to the distribution $\mathcal{P}$ of $x$'s, and by doing rejection sampling until we find a point with $(x, z)$ with $z \in S$. To bound the time needed for this step, we need to understand the mass of the survival set $S$ for some $x \sim \mathcal{P}$ and some vector of parameters $(v, \lambda) \in D$. See Supplementary Material B.2.

(c) We show in the supplementary material that $D$ is a convex set and define an efficient algorithm to project our estimates to $D$. It is elaborated in Supplementary Material A.2

For more details on the way we solve the computational problems (a) - (c) we refer to the supplementary material.

## 4.3 Bounded Step Variance, Bounded Domain, and Strong Convexity

In this section we present the statements that establish the properties (i), (ii), (iv) that we need in order to apply Theorem 3.4 and Theorem 3.5.

**Theorem 4.1.** *For every* $(v, \lambda) \in D$, *we have*

$$\mathbb{E}\left[\|(y - z)x\|_2^2\right] \leq \mathrm{poly}(1/a)(1 + \beta^2 + \sigma_0^4), \quad \mathbf{H}(v, \lambda) \succeq b \cdot \exp\left(-\mathrm{poly}(1/a)(1 + \frac{\beta}{\sigma_0^2})\right)\begin{pmatrix} \sigma_0^2 I & 0 \\ 0 & \sigma_0^4 \end{pmatrix}$$

*where,* $x \sim \mathcal{P}$, $y \sim F_x$ *and* $z \sim Q_x$ *and* $\mathbf{H}$ *from* (4.3).

The proof of Theorem 4.1 can be found in Supplementary Material B.3. Also, it is easy to see that the diameter of $D$ is bounded: $\|v\|_2 < \|w\|_2/\sigma^2 \leq 8(5 - 2\log a)\sqrt{\beta}/\sigma_0^2$ and $|\lambda| < 1/\sigma^2 \leq 8(5 - 2\log a)/\sigma_0^2$

## 4.4 Feasibility of Optimal Solution

We need to calculate the probability for the optimal point to be in the projection set, i.e., $w^*, \sigma^* \in D$.

**Lemma 4.2.** *Under Assumptions 2.3 and 2.2 we have that,* $\mathbb{P}((w^*, \sigma^*) \in D) \geq 13/16$.

The proof of this Lemma can be found in Supplementary Material B.4.

## 4.5 Proof of Theorem 3.2

The algorithm that proves Theorem 3.2 starts with splitting the samples $n$ into three parts of equal size in order to avoid dependence. The first part of $n/3$ samples is used to compute the OLS estimates

$w_0$, where $w_0$ is used in the initialization step. The second part of $n/3$ samples is used to compute the OLS estimate of $\sigma_0$, where $\sigma_0$ is used both to initialize and define the projection set $D$, and is used at every step where a projection to $D$ is needed. The number of sample to define $D$ is $\mathrm{poly}(\frac{\sigma^*,\beta}{a,b}, \frac{1}{\sigma^*})$, and our $n$ itself will reach this bound (see Section B.4.) Finally, the third part of the $n/3$ samples is used in our main PSGD algorithm. For the rest of the proof we abuse the notation and we use $n \leftarrow n/3$ for simplicity. The next thing that we need to decide is the learning rates that we are going to use for the PSGD. We will analyze PSGD under two possible choices of its learning rate schedule, namely $\eta_t = c/\sqrt{t}$ and $\eta_t = 1/(\zeta \cdot t)$, and will show that it produces estimates for the ground-truth parameters that scale as $A_1 = \frac{\mathrm{poly}(\frac{\sigma_0 \cdot \beta}{a \cdot b}, \frac{1}{a \cdot \sigma_0}) \cdot \log(n)}{n^{1/4}}$ and $A_2 = \frac{\mathrm{poly}(\sigma_0) \cdot \exp\left(\mathrm{poly}(\frac{\beta}{a \cdot \sigma_0})\right)}{b\sqrt{n}}$ respectively, for some polynomials that we will make explicit in Section B.7.

**Analysis for learning schedule $\eta_t = c/\sqrt{t}$.** First, by combining Theorem 3.4 and Theorem 4.1, and noticing the relationship between $\sigma_0$ and $\sigma^*$ in the projection set, we get the following corollary whose proof can be found in Section B.7.

**Corollary 4.3.** *Suppose that* $(v^*, \lambda^*) = (w^*/\sigma^{*2}, 1/\sigma^{*2}) \in D$ *and* $\bar{\ell}$ *is defined as in* (4.2). *Then, the output* $(\hat{v}, \hat{\lambda})$ *of Algorithm 1 satisfies* $\mathbb{E}\left[\bar{\ell}(\hat{v}, \hat{\lambda}) - \bar{\ell}(v^*, \lambda^*)\right] \leq \frac{\mathrm{poly}(\frac{\sigma^* \cdot \beta}{a \cdot b}, \frac{1}{\sigma^*}) \cdot \log(n)}{n^{1/2}}$.

Our next step is utilize the above bound to so that our estimates achieve small errors in the parameter space. For this we use the strong convexity of $\bar{\ell}$ at the optimum.

**Lemma 4.4.** *The Hessian at the true* $(v^*, \lambda^*)$ *satisfies* $\mathbf{H}(v^*, \lambda^*) \succeq \frac{b \times \mathrm{poly}(a)}{\mathrm{poly}(1/a, \beta/\sigma^{*2})} \begin{pmatrix} \sigma^{*2} I & 0 \\ 0 & \sigma^{*4} \end{pmatrix}$.

The proof of Lemma 4.4 can be found in Supplementary Material B.3.1. We also utilize the following lemma to calculate an upper bound on $\bar{\ell}(v^*, \lambda^*)$:

**Lemma 4.5.** (Lemma 3 in Daskalakis et al. (2019)) *Let* $x, w \in \mathbb{R}^k$, $\sigma > 0$, *then* $\mathbb{E}_{y \sim \mathcal{N}(w^T x, \sigma^2, S)}(y - w^T x)^2 \leq (4 - 2 \log \alpha(w, \sigma, x))\sigma^2$.

By the above lemma, we have:

$$\bar{\ell}(v^*, \lambda^*) = \mathop{\mathbb{E}}_{x \in \mathcal{P}} \mathop{\mathbb{E}}_{y} \left[ \frac{(y - w^{*T} x)^2}{2\sigma^{*2}} + \log \alpha(w^*, \sigma^*, x) \right] \leq 4 - 2 \log \min_{x \in \mathcal{P}} \alpha(w^*, \sigma^*, x) \leq 4 - 2 \log a$$

Applying Markov's inequality we have

$$\mathbb{P}\left(\bar{\ell}(\hat{v}, \hat{\lambda}) - \bar{\ell}(v^*, \lambda^*) > 7\mathbb{E}\left[\bar{\ell}(\hat{v}, \hat{\lambda}) - \bar{\ell}(v^*, \lambda^*)\right]\right) < \frac{1}{7}. \tag{4.4}$$

The inequality implies that, with at least $6/7$ probability, the actual difference is at most 7 times its expectation. Also, the probability that $(v^*, \lambda^*) \in D$ is at least $13/16$ by Lemma 4.2. Combining these two, we have at least $13/16 - 1/7 > 2/3$ probability to hold:

$$\bar{\ell}(\hat{v}, \hat{\lambda}) - \bar{\ell}(v^*, \lambda^*) < \frac{\mathrm{poly}(\frac{\sigma^* \cdot \beta}{ab}, \frac{1}{\sigma^*}) \cdot \log(n)}{n^{1/2}} \tag{4.5}$$

Now applying Lemma 4.4, since $\hat{v}, \hat{\lambda}$ converge to $v^*, \lambda^*$, when they have small distance, we have

$$\|\hat{v} - v^*\|_2^2 + (\hat{\lambda} - \lambda^*)^2 \leq \frac{2}{\delta}\left(\bar{\ell}(\hat{v}, \hat{\lambda}) - \bar{\ell}(v^*, \lambda^*)\right) \leq \frac{\mathrm{poly}(\frac{\sigma^* \cdot \beta}{ab}, \frac{1}{\sigma^*}) \cdot \log(n)}{\delta \cdot n^{1/2}} \tag{4.6}$$

where $\mathbf{H}(v^*, \lambda^*) \succeq \delta I$, and $\delta^{-1} = \mathrm{poly}\left(\frac{\sigma^* \cdot \beta}{a \cdot b}, \frac{1}{\sigma^*}\right)$.

Finally, doing some simple calculations for the reparameterization —see Section B.7 in supplementary material— we have that

$$\|\hat{w} - w^*\|_2 + \left|\hat{\sigma}^2 - \sigma^{*2}\right| \leq \sqrt{2\left(\|\hat{w} - w^*\|_2^2 + \left|\hat{\sigma}^2 - \sigma^{*2}\right|^2\right)} \leq \mathrm{poly}(1/a, \sigma^*) \left\|(\hat{v}, \hat{\lambda}) - (v^*, \lambda^*)\right\|_2. \tag{4.7}$$

Combining inequalities (4.4), (4.5), (4.6), (4.7), inequality (3.1) of Theorem 3.2 follows.

**Analysis for learning schedule $\eta_t = 1/(\zeta t)$.** We combine Theorem 3.5 and Theorem 4.1 to get the following corollary whose proof can be found in Section B.6.

**Corollary 4.6.** *Suppose the true parameters $(v^*, \lambda^*) = (w^*/\sigma^{*2}, 1/\sigma^{*2}) \in D$ and $\bar{\ell}$ is defined as equation* (4.2). *Then, for $\hat{v}, \hat{\lambda}$ being the output of Algorithm 1, it holds that*

$$\mathbb{E}\left[\left\|(\hat{v}, \hat{\lambda}) - (v^*, \lambda^*)\right\|_2^2\right] \leq \text{poly}\left(\beta, \sigma^* + \frac{1}{\sigma^*}, \frac{1}{a}\right) \times \exp\left(\text{poly}(1/a)\left(1 + \frac{\beta}{\sigma^{*2}}\right)\right)\frac{1}{b^2 n}.$$

Similarly —details can be found in Section B.7 of the supplementary material— to those used to prove inequality (3.1) above, we get that with probability $\geq 2/3$ it holds that $\left\|(\hat{v}, \hat{\lambda}) - (v^*, \lambda^*)\right\|_2^2 \leq \frac{\text{poly}(\sigma^*) \cdot \exp\left(\text{poly}(\frac{\beta}{a \cdot (\sigma^*)})\right)}{b^2 \cdot n}$. Then, we can prove (3.2) using the definition of the projection set $D$.

## 5  Inference and Confidence Regions

We now prove the asymptotic normality of our estimates (Theorem 5.1), which we use to obtain their confidence regions (Corollary 5.2). For the proofs we refer to the Section B.8.

**Theorem 5.1.** *Let $(\hat{w}, \hat{\sigma}^2)$ be the output of Algorithm 1 with $\eta_t = 1/(\zeta \cdot t)$. Then, the vector $\sqrt{n}S^{-1/2}(\binom{\hat{w}}{\hat{\sigma}^2}) - \binom{w^*}{\sigma^{*2}})$, asymptotically converges to $\mathcal{N}(0, I_{k+1})$, where*

- $S = \frac{1}{\zeta}R^T \Sigma R,\ R = \begin{pmatrix} 1/\hat{\sigma}^2 I & -\hat{w}/\hat{\sigma}^4 \\ 0 & -1/\hat{\sigma}^4 \end{pmatrix}$,

- $\Sigma$ *is the matrix that satisfies* $\left(\hat{\mathbf{H}} - \frac{\zeta}{2}I\right)\Sigma + \Sigma\left(\hat{\mathbf{H}} - \frac{\zeta}{2}I\right) = \Gamma(\hat{v}, \hat{\lambda}) = \Gamma\left(\frac{\hat{w}}{\hat{\sigma}^2}, \frac{1}{\hat{\sigma}^2}\right)$,

- $\hat{\mathbf{H}}$ *is the empirical estimation of the Hessian matrix defined by* (4.3) *at the point* $(\hat{v}, \hat{\lambda})$,

- $\Gamma(v, \lambda) = \frac{1}{n}\sum_{i=1}^n \frac{\partial l(v,\lambda; x^{(i)}, y^{(i)})}{\partial(v,\lambda)} \frac{\partial l(v,\lambda; x^{(i)}, y^{(i)})}{\partial(v,\lambda)}^T$.

**Corollary 5.2.** *Let $q_\alpha$ be the $1 - \alpha$ quantile of distribution $\chi_{k+1}^2$. Then the $1 - \alpha$ asymptotic confidence region of $(w, \sigma^2)$ is* $\left(\binom{w}{\sigma^2} - \binom{\hat{w}}{\hat{\sigma}^2}\right)^T R^T \Sigma R \left(\binom{w}{\sigma^2} - \binom{\hat{w}}{\hat{\sigma}^2}\right) \leq \frac{q_\alpha}{\zeta \cdot n}$.

## 6  Synthetic Experiments

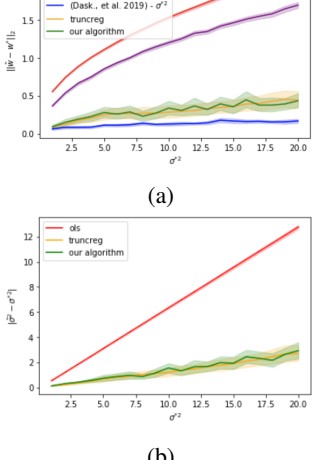

(a)

(b)

Figure 1: Figure (a) shows $\hat{w}$ estimation errors and Figure (b) shows the $\hat{\sigma}^{*2}$ estimation errors for each $\sigma^{*2}$.

In Figure 1, we show how our algorithm performs against models with large noise variances. For this experiment, we sample a 2 dimensional ground-truth weight vector $w^* \overset{i.i.d.}{\sim} \mathcal{U}(-1, 1)$ and generate 1000 $(x^{(i)}, y^{(i)})$ pairs according to $x^{(i)} \overset{i.i.d.}{\sim} \mathcal{U}(-5, 5)$ and $y^{(i)} = w^{*T}x^{(i)} + \varepsilon^{(i)}$; where $\varepsilon^{(i)} \sim \mathcal{N}(0, \sigma^{*2})$. We fix $w^*$ and $x^{(0)}, ..., x^{(n)}$ across all trials, but vary $\sigma^{*2}$. After adding Gaussian noise to our ground-truth predictions, we left truncate at zero, removing all of the pairs who's $y^{(i)}$ is negative; retaining approximately 50% of the original samples. We vary $\sigma^{*2}$ over the interval $[1, 20]$ and evaluate how well our procedure recovers $w^*$ and $\sigma^{*2}$ in comparison to OLS, Daskalakis et al. (2019), assuming $\sigma_0^2$ is $\sigma^{*2}$, Daskalakis et al. (2019), given $\sigma^{*2}$, and *truncreg*, an R package for truncated regression. We acknowledge that when the noise variance is known, Daskalakis et al. (2019) performs the best, but both our algorithm and *truncreg* perform comparably under this simple truncation. We emphasize though that *truncreg* cannot be applied in settings where the truncation is more complicated that a simple interval whereas our method can still be applied. Also notice that running Daskalakis et al. (2019) with an incorrect $\sigma^{*2}$ produces erroneous results.

In Figure 2, we show that our theoretical error bounds hold in practice. For this experiment, we set $w^*$ to a ten dimensional weight

vector of ones, sample $n$ $(x^{(i)}, y^{(i)})$ pairs, where $x^{(i)} \overset{i.i.d.}{\sim} \mathcal{N}(0, I)$ and $\varepsilon^{(i)} \overset{i.i.d}{\sim} \mathcal{N}(0, 1)$, and right truncate at zero. We generate our data once. However, for each experiment, we randomly select a $k$ samples that exceed zero, and run our experiment. We vary $k$ over the interval $[10, 5000]$. Our results show that the error bound asymptotically approaches zero at a rate that reaffirms our theoretical analysis.

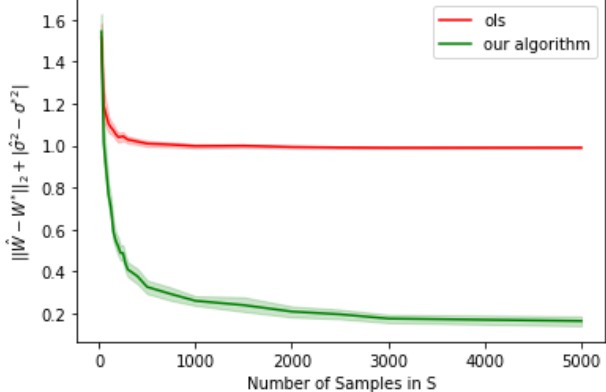

Figure 2: Comparison of the proposed method with ordinary least squares.

# 7   Semi-Synthetic Experiment

Here we show results for an additional experiment that we conducted on semi-synthetic data. For this experiment, we used the Aldrin (2004) dataset. The Aldrin (2004) dataset was originally collected by the Norwegian Public Roads Administration for a study of air pollution at a road in Oslo, Norway. The dataset consists of 500 observations. Interestingly enough, it is common for environmental data to be truncated because of problems in reliably measuring low concentrations.

For this experiment, we apply left truncation to the dataset, varying a truncation parameter $C$ over the interval $[1.0, 4.0]$. We run our procedure a total of 10 times and retain the trial that has the smallest gradient as our algorithm's prediction. We conduct the experiment with the same hyperparameters as for the synthetic data experiments. Below, we report our results.

In Figure 3 (a), we show the $L^2$ distance between the model parameters of each method and the ground-truth, which we take to be the OLS estimation without truncation. In this figure, we show that our method performs very comparably to Croissant & Zeileis (2018). We also note that, at high values of the truncation, both methods exhibit non-monotonicity in their estimation error, which is presumably due to model misspecification and the truncation being too aggressive.

In Figure 3 (b), we report the $L^1$ distance between the predicted noise variance and the ground-truth noise variance. Again, our method performs very comparably to Croissant & Zeileis (2018), and we observe non-monotonicity at high values of the truncation.

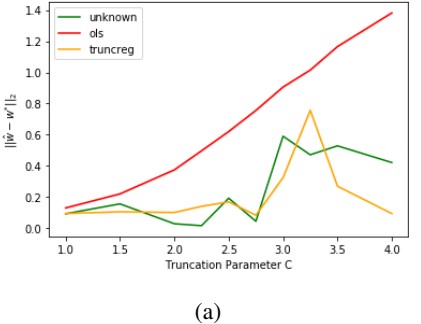

(a)

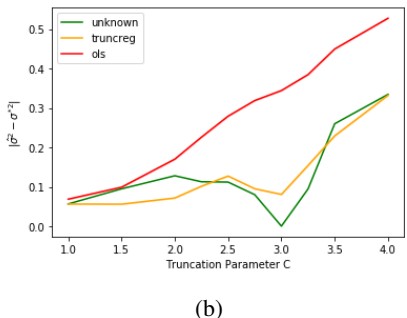

(b)

Figure 3: Our method's performance on the semi-synthetic PM10 dataset.

## Acknowledgements

CD and PS were supported by NSF Awards CCF-1901292, DMS-2022448 (FODSI) and DMS-2134108, by a Simons Investigator Award, by the Simons Collaboration on the Theory of Algorithmic Fairness, by a DSTA grant, and by the DOE PhILMs project (No. DE-AC05-76RL01830). MZ was supported by NSF DMS-2023505 (FODSI).

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
