# A  Projection Set

If it is not specified, all the norms for vectors are $L^2$ norms, by default.

## A.1  Proof for the convexity of the projection set

Recall that the projection set is

$$D_r = \left\{ (v, \lambda) = \left( \frac{w}{\sigma^2}, \frac{1}{\sigma^2} \right) \in \mathbb{R}^k \times \mathbb{R} \mid \frac{1}{8(5 - 2\log a)} \leq \frac{\sigma^2}{\sigma_0^2} \leq \frac{96}{a^2}, \|w\|_2^2 \leq \beta \right\}$$

So we have

$$\|w\|^2 = w^T w = (v/\lambda)^T (v/\lambda) = v^T v/\lambda^2 = \|v\|^2/\lambda^2$$

Therefore, if we transform it into a direct representation with $v, \lambda$, we get

$$D_r = \left\{ (v, \lambda) \in \mathbb{R}^k \times \mathbb{R}_{>0} \mid \frac{a^2}{96\sigma_0^2} \leq \lambda \leq \frac{8(5 - 2\log a)}{\sigma_0^2}, \|v\|^2 \leq \beta \lambda^2 \right\}$$

Where $\mathbb{R}_{>0}$ denotes the set of positive real numbers. We prove that this is a convex set by showing that it is an (infinite) intersection of convex sets.

The first constraint $\frac{a^2}{96\sigma_0^2} \leq \lambda \leq \frac{8(5-2\log a)}{\sigma_0^2}$ is a region between two hyper planes, which is also a convex constraint.

We prove that the second constraint $\|v\|^2 \leq \beta \lambda^2$ is convex. Suppose we have two vectors such that $\|v_1\|^2 \leq \beta \lambda_1^2$ and $\|v_2\|^2 \leq \beta \lambda_2^2$. We need to prove that $\|\mu v_1 + (1 - \mu)v_2\|^2 \leq \beta(\mu \lambda_1 + (1 - \mu)\lambda_2)^2$ for all $\mu \in (0, 1)$. With this condition and knowing that $\lambda_1, \lambda_2 > 0$, we know that $\|v_1\| \leq \sqrt{\beta}\lambda_1$ and $\|v_2\| \leq \sqrt{\beta}\lambda_2$. By the triangle inequality, we have $\|\mu v_1 + (1 - \mu)v_2\|^2 \leq (\mu\|v_1\| + (1-\mu)\|v_2\|)^2 \leq (\mu\sqrt{\beta}\lambda_1 + (1-\mu)\sqrt{\beta}\lambda_2)^2 = \beta(\mu\lambda_1 + (1-\mu)\lambda_2)^2$. So, we proved the convexity of the third constraint.

Since the three constraints are all convex and the projection set is the intersection of the three constraints, the projection set is convex.

## A.2  Algorithm for projecting to the projection set

We have an explicit formula for projecting to the projection set. For convenience, we denote $\lambda_{\min} = \frac{a^2}{96\sigma_0^2}$ and $\lambda_{\max} = \frac{8(5-2\log a)}{\sigma_0^2}$. Thus, the projection formula projects $(v, \lambda) \in D_r$ by minimizing $\|(v_0, \lambda_0) - (v, \lambda)\|$ as below:

$$\begin{cases} (v_0, \lambda_0) & \lambda_{\min} \leq \lambda_0 \leq \lambda_{\max}, \|v_0\|^2 \leq \beta\lambda_0^2 \\ (v_0, \lambda_{\max}) & \lambda_0 \geq \lambda_{\max}, \|v_0\|^2 \leq \beta\lambda_{\max}^2 \\ \left( \frac{\sqrt{\beta}\lambda_{\max}}{\|v_0\|} v_0, \lambda_{\max} \right) & \lambda_0 \geq \lambda_{\max}, \|v_0\|^2 \geq \beta\lambda_{\max}^2 \\ (v_0, \lambda_{\min}) & \lambda_0 \leq \lambda_{\min}, \|v_0\|^2 \leq \beta\lambda_{\min}^2 \\ \left( \frac{\sqrt{\beta}\lambda_{\min}}{\|v_0\|} v_0, \lambda_{\min} \right) & \lambda_0 \leq \lambda_{\min}, \sqrt{\beta}\lambda_{\min} \leq \|v_0\| \leq \sqrt{\beta}\lambda_{\min} + \frac{\lambda_{\min} - \lambda_0}{\sqrt{\beta}} \\ \left( \frac{\sqrt{\beta}\lambda_{\max}}{\|v_0\|} v_0, \lambda_{\max} \right) & \lambda_0 \leq \lambda_{\max}, \|v_0\| \geq \sqrt{\beta}\lambda_{\max} + \frac{\lambda_{\max} - \lambda_0}{\sqrt{\beta}} \\ \left( \frac{\beta\|v_0\| + \sqrt{\beta}\lambda_0}{(\beta + 1)\|v_0\|} v_0, \frac{\sqrt{\beta}\|v_0\| + \lambda_0}{\beta + 1} \right) & \text{Otherwise} \end{cases}$$

# B  Missing Proofs

If it is not specified, all the norms for vectors are $L^2$ norms, by default.

## B.1  Sampling the Gradient of the Objective Function

Let $\mathcal{D}^*$ be the joint distribution of the observed pairs $(x, y)$, where $(w^*, \sigma^*)$ are the ground truth parameters. Notice that we have a sampler that generates samples $(x, y)$ from $D^*$. Also let's define $\mathcal{D}_{(w,\sigma)}$ as the joint distribution of the pairs $(x, y)$, when the vector of parameters is $(w, \sigma)$. We recall that $\mathcal{P}$ is the marginal distribution of $x$ of the joint distribution $\mathcal{D}^*$. Also, $\mathcal{P}_0$ is the original distribution of $x$ before truncation.

So, after truncation, we have

$$\mathcal{D}^*(\bar{x}, \bar{y}) = \frac{\mathbf{1}\{\bar{y} \in S\} \cdot \mathcal{N}(w^{*T}\bar{x}, \sigma^{*2}; \bar{y}) \cdot \mathcal{P}_0(\bar{x})}{\mathbb{E}_{x \sim \mathcal{P}_0}\left[\alpha(w^*, \sigma^*, \bar{x}; S))\right]}$$

We note that $\mathcal{N}(w^{*T}\bar{x}, \sigma^{*2}; \bar{y})$ is the probability density of $\bar{y}$; that is, $\exp\left(-\frac{(\bar{y}-w^{*T}x)^2}{2\sigma^{*2}}\right)$. To make it explicit, let's say that the relationship between $\mathcal{P}$ and $\mathcal{P}_0$ is,

$$\mathcal{P}(\bar{x}) = \int_{\mathbb{R}} \frac{\mathbf{1}\{\bar{y} \in S\} \cdot \mathcal{N}(w^{*T}\bar{x}, \sigma^{*2}; \bar{y}) \cdot \mathcal{P}_0(\bar{x})}{\mathbb{E}_{x \sim \mathcal{P}_0}\left[\alpha(w^*, \sigma^*, \bar{x}; S)\right]} d\bar{y} = \frac{\alpha(w^*, \sigma^*, \bar{x}; S)}{\mathbb{E}_{x \sim \mathcal{P}_0}\left[\alpha(w^*, \sigma^*, \bar{x}; S))\right]} \cdot \mathcal{P}_0(\bar{x})$$

Now we can sample the gradient. The gradient is

$$\frac{\partial \bar{\ell}}{\partial v} = \mathbb{E}_{x \sim \mathcal{P}}\left[\mathbb{E}_{z \sim Q_x}[z \cdot x] - \mathbb{E}_{y \sim F_x}[y \cdot x]\right] \quad \frac{\partial \bar{\ell}}{\partial \lambda} = \frac{1}{2} \mathbb{E}_{x \sim \mathcal{P}}\left[\mathbb{E}_{y \sim F_x}[y^2] - \mathbb{E}_{z \sim Q_x}[z^2]\right],$$

Where $F_x = \mathcal{N}(w^{*T}x, \sigma^{*2}, S)$ and $Q_x = \mathcal{N}(w^T x, \sigma^2, S)$. Further, note that

$$\mathbb{E}_{x \sim \mathcal{P}}\left[\mathbb{E}_{y \sim \mathcal{N}(w^{*T}x, \sigma^{*2}, S)}[y \cdot x]\right]$$

$$= \int_x \int_S (y \cdot x) \frac{\mathcal{N}(w^{*T}x, \sigma^{*2}; \bar{y})}{\mathcal{N}(w^{*T}x, \sigma^{*2}; S)} \cdot dy \cdot \mathcal{P}(x) \cdot dx$$

$$= \int_x \int_S (y \cdot x) \frac{\mathcal{N}(w^{*T}x, \sigma^{*2}; \bar{y})}{\mathcal{N}(w^{*T}x, \sigma^{*2}; S)} \cdot dy \cdot \frac{\mathcal{N}(w^{*T}\bar{x}, \sigma^{*2}; S)}{\mathbb{E}_{x \sim \mathcal{P}_0}\left[\mathcal{N}(w^{*T}x, \sigma^{*2}; S)\right]} \cdot \mathcal{P}_0(\bar{x}) \cdot dx$$

$$= \int_x \int_S (y \cdot x) \frac{\mathcal{N}(w^{*T}x, \sigma^{*2}; \bar{y})}{\mathbb{E}_{x \sim \mathcal{P}_0}\left[\mathcal{N}(w^{*T}x, \sigma^{*2}; S)\right]} \cdot \mathcal{P}_0(\bar{x}) \cdot dy \cdot dx = \mathbb{E}_{(x,y) \sim \mathcal{D}^*}[y \cdot x]$$

At each gradient step, we sample a pair of $x, y$ and multiply them together. Similarly we sample $\frac{1}{2}y^2$, where $x \sim \mathcal{P}$ and $y \sim F_x$, and ignore $x$ after sampling. For the second term, we use rejection sampling to generate the data $z \cdot x$ for $z \sim Q_x$ and $x \sim \mathcal{P}$. Here, since we know the $w$ and $\sigma$, the generation process is simple. First, we generate $x$ using $D^*$ (we discard the $y$ value) and then sample $z = w^T x + \varepsilon$ where $\varepsilon = \mathcal{N}(0, \sigma^2)$. If $z \notin S$, we continue to sample $\varepsilon$ until we get $z \in S$. Similarly, we can sample $\frac{1}{2}z^2$ where $x \sim \mathcal{P}, z \sim Q_x$ in this way, and we can ignore $x$ after sampling.

Notice that if $w, \sigma = w^*, \sigma^*$, we have $Q_x = F_x$, so

$$\frac{\partial \bar{\ell}}{\partial v}(w^{*T}, \sigma^{*2}) = \mathbb{E}_{x \sim \mathcal{P}}\left[\mathbb{E}_{z \sim Q_x}[z \cdot x] - \mathbb{E}_{y \sim F_x}[y \cdot x]\right] = 0$$

and

$$\frac{\partial \bar{\ell}}{\partial \lambda} = \frac{1}{2} \mathbb{E}_{x \sim \mathcal{P}}\left[\mathbb{E}_{y \sim F_x}[y^2] - \mathbb{E}_{z \sim Q_x}[z^2]\right],$$

## B.2 Auxilliary Lemmas for Survival Probability of Feasible Points

For many of the following proofs, we require an estimation of $\alpha(w, \sigma, x)$ for a feasible $w$.

**Lemma B.1.** *Let* $x, w, w' \in \mathbb{R}^k$ *and* $\sigma, \sigma' > 0$. *Then,*

$$\log\left(\frac{1}{\alpha(w, \sigma, x)}\right) \leq \max\left(1, \frac{2\sigma'^2}{\sigma^2}\right) \log\frac{1}{\alpha(w', \sigma', x)} + \max\left(0, \frac{4\sigma'^2}{\sigma^2} - 2\right) + \frac{2\sigma'^2}{\sigma^2}\left(\frac{(w^T x - w'^T x)^2}{2\sigma'^2}\right)$$

*Proof.* Now, let's define $D$, $D'$, and $D'_S$ as $D = \mathcal{N}(w^T x, \sigma)$, $D' = \mathcal{N}(w'^T x, \sigma')$, and $D'_S = \mathcal{N}(w'^T x, \sigma', S)$. Then, we have:

$$\alpha(w, \sigma, x) = \mathbb{E}_{y \sim D}\left[\mathbf{1}_{y \in S}\right] = \mathbb{E}_{y \sim D'}\left[\mathbf{1}_{y \in S} \exp\left(\frac{(y - w'^T x)^2}{2\sigma'^2} - \frac{(y - w^T x)^2}{2\sigma^2}\right)\right]$$

$$= \alpha(w', \sigma', x) \cdot \mathbb{E}_{y \sim D'_S}\left[\exp\left(\frac{(y - w'^T x)^2}{2\sigma'^2} - \frac{(y - w^T x)^2}{2\sigma^2}\right)\right]$$

$$\geq \alpha(w', \sigma', x) \cdot \exp\left(\mathbb{E}_{y \sim D'_S}\left[\frac{(y - w'^T x)^2}{2\sigma'^2} - \frac{(y - w^T x)^2}{2\sigma^2}\right]\right).$$

Now, from Cauchy-Schwarz we have that

$$\frac{(y - w^T x)^2}{2\sigma^2} \leq \frac{2\sigma'^2}{\sigma^2} \left( \frac{(y - w'^T x)^2}{2\sigma'^2} + \frac{(w^T x - w'^T x)^2}{2\sigma'^2} \right)$$

and if we apply this to the above expression and take the logarithm of both sides of the inequality we get

$\log \alpha(w, \sigma, x)$

$$\geq \log \alpha(w', \sigma', x) + \mathop{\mathbb{E}}_{y \sim D_S'} \left[ \left( 1 - \frac{2\sigma'^2}{\sigma^2} \right) \frac{(y - w'^T x)^2}{2\sigma'^2} \right] - \mathop{\mathbb{E}}_{y \sim D_S'} \left[ \frac{2\sigma'^2}{\sigma^2} \left( \frac{(w^T x - w'^T x)^2}{2\sigma'^2} \right) \right]$$

$$\geq \log \alpha(w', \sigma', x) + \mathop{\mathbb{E}}_{y \sim D_S'} \left[ \min\left( 0, 1 - \frac{2\sigma'^2}{\sigma^2} \right) (2 - \log \alpha(w', \sigma', x)) - \frac{2\sigma'^2}{\sigma^2} \left( \frac{(w^T x - w'^T x)^2}{2\sigma'^2} \right) \right]$$

$$= \min\left( -1, -\frac{2\sigma'^2}{\sigma^2} \right) (-\log \alpha(w', \sigma', x)) - \min\left( 0, 2 - \frac{4\sigma'^2}{\sigma^2} \right) - \frac{2\sigma'^2}{\sigma^2} \left( \frac{(w^T x - w'^T x)^2}{2\sigma'^2} \right)$$

The second to last line is followed from Lemma 4.5 and the Lemma follows. □

## B.3   Proof of Theorem 4.1 and Lemma 4.4

Denote $s = \frac{\max_{\sigma \in D} \sigma}{\min_{\sigma \in D} \sigma} = \sqrt{8(5 - 2\log a)\frac{96}{a^2}}$ be the largest possible difference between variances, where $D$ is the projection set defined in 3.1.

Next, to prove our strong convexity result, we use the following anti-concentration bound.

**Theorem B.2** (Theorem 8 in Carbery & Wright (2001b)). *There is an absolute constant $\mathcal{C}$ such that if $p : \mathbb{R}^n \to \mathbb{R}$ is a polynomial of degree at most $d$, $0 < q < \infty$, and $\mu$ is a log-concave probability measure on $\mathbb{R}^n$, then*

$$\left( \int |p(x)|^{q/d} d\mu(x) \right)^{1/q} \alpha^{-1/d} \cdot \mu \left\{ x \in \mathbb{R}^n : |p(x)| \leq \alpha \right\} \leq \mathcal{C}q$$

We also need the following lemma for bounding the survival probability for any parameters in $D$.

**Lemma B.3.** *For $w, \sigma \in D$, we can find a lower bound for the survival probability: $\frac{1}{\alpha(w, \sigma, x^{(i)})} \leq \exp(\text{poly}(1/a)(1 + \frac{\beta}{\sigma_0^2}))$*

*Proof.* From Lemma B.1, when we plug $\sigma' = \sigma^*$, $w' = w^*$, we can derive that

$$\log\left( \frac{1}{\alpha(w, \sigma, x)} \right) \leq \max\left( 1, \frac{2\sigma^{*2}}{\sigma^2} \right) \log\left( \frac{1}{\alpha(w^*, \sigma^*, x)} \right)$$

$$+ \max\left( 0, \frac{4\sigma^{*2}}{\sigma^2} - 2 \right) + \frac{2\sigma^{*2}}{\sigma^2} \left( \frac{(w^T x - w^{*T} x)^2}{2\sigma^{*2}} \right)$$

Since $\sigma, \sigma^* \in D$, we have $\frac{\sigma^{*2}}{\sigma^2} \le s^2$. Thus, we have

$$\log\left(\frac{1}{\alpha(w,\sigma,x)}\right)$$

$$\le \max\left(1, \frac{2\sigma^{*2}}{\sigma^2}\right)\log\left(\frac{1}{\alpha(w^*,\sigma^*,x)}\right) + \max\left(0, \frac{4\sigma^{*2}}{\sigma^2} - 2\right) + \frac{2\sigma^{*2}}{\sigma^2}\left(\frac{(w^Tx - w^{*T}x)^2}{2\sigma^{*2}}\right)$$

$$\le 2s^2\left(\log\frac{1}{\alpha(w^*,\sigma^*,x)} + 2 + \left(\frac{(w^Tx - w^{*T}x)^2}{2\sigma^{*2}}\right)\right)$$

$$\le 2s^2\left(-\log a + 2 + \left(\frac{(\|w\|^2 + \|w^*\|^2)\|x\|^2}{\sigma^{*2}}\right)\right)$$

$$\le 2s^2\left(-\log a + 2 + \left(\frac{2\beta}{\sigma^{*2}}\right)\right)$$

Since $\sigma_0$ and $\sigma^*$ only have a polynomial difference, we have $\log\left(\frac{1}{\alpha(w,\sigma,x)}\right) \le \text{poly}(1/a)(1 + \frac{\beta}{\sigma_0^2})$ which finishes the proof. □

**Lemma B.4.** *Let $z \sim \mathcal{N}(\mu, \sigma^2, S)$ be a truncated normal variable. Assume $\alpha(\mu, \sigma^2, S) = a$. Then, we have that*

$$\text{Var}(z^2) = 2\mu^2\sigma^2(O(\log(a))) + \sigma^4 O(\log(a)^2)$$

*Proof.* Denote an affine transformation $S' = \{\frac{x-\mu}{\sigma} | x \in S\}$. So $D = \mathcal{N}(\mu, \sigma, S)$ is transformed into $D' = \mathcal{N}(0, 1, S')$, while the survival probability is maintained. We then have

$$\text{Var}_{z\sim D}(z^2) = \text{Var}_{y\sim D'}((\mu + \sigma y)^2) = \text{Var}_{y\sim D'}(2\sigma\mu y + \sigma^2 y^2)$$
$$\le 8(\sigma\mu)^2\text{Var}_{y\sim D'}(y) + 2(\sigma)^4\text{Var}_{y\sim D'}(y^2)$$

By Lemma 4.5 we have $\text{Var}_{y\sim D'}[y] \le \mathbb{E}[y^2] \le 4 - 2\log a$, and

$$\text{Var}_{D'}(y^2) \le \mathbb{E}_{D'}(y^4) \le \frac{\int_q^\infty x^4 e^{-\frac{x^2}{2}}\,dx}{\int_q^\infty e^{-\frac{x^2}{2}}\,dx} = \frac{\int_q^\infty x^4 e^{-\frac{x^2}{2}}\,dx}{a/2}$$

where $q$ satisfies $\int_q^\infty e^{-\frac{x^2}{2}}\,dx = a/2$.

Now, all we need to prove is that $q = O(\sqrt{-\log a})$. Notice that if $q > 2\sqrt{-\log a}$ (in the case $a < \sqrt{2}/2$ and $q > 1$), then we have

$$a/2 = \int_q^\infty e^{-z^2/2}dz = \int_{q^2/2}^\infty \frac{1}{\sqrt{z}}e^{-z}dz < \frac{\sqrt{2}}{q}e^{-q^2/2} < \sqrt{2}a^2/q < a$$

This is a contradiction.

By integration by parts, we obtain

$$\int_q^\infty z^4 e^{-z^2/2}dz = q^3 e^{-q^2/2} + 3\int_q^\infty z^2 e^{-z^2/2}dz = q^3 e^{-q^2/2} + 3qe^{-q^2/2} + 3\int_q^\infty e^{-z^2/2}dz$$

and

$$\frac{\int_q^\infty z^4 e^{-z^2/2}dz}{\int_q^\infty e^{-z^2/2}dz} = 3 + \frac{q^3 e^{-q^2/2} + 3qe^{-q^2/2}}{\int_q^\infty e^{-z^2/2}dz} \le 3 + \frac{q^3 e^{-q^2/2} + 3qe^{-q^2/2}}{\int_q^{q+1/q} e^{-(q^2+3)/2}dz} \le 3 + 15q^2 + 5q^4$$

Since when $a < \sqrt{2}/2$, we have both $q < 2\sqrt{-\log a}$ and $\text{Var}_{Ds'}(y^2) = O(\log a^2)$. The lemma follows. □

### B.3.1 Strong Convexity

Now, let's deal with the Hessian Matrix.

First, we consider the case without truncation, that is,

$$\mathbb{E}_{\mathcal{P}}\left[\begin{pmatrix} \mathrm{Var}_x[z]xx^T & -\mathrm{Cov}_x\left[\frac{1}{2}z^2, z\right]x \\ -\mathrm{Cov}_x\left[\frac{1}{2}z^2, z\right]x^T & \mathrm{Var}_x\left[\frac{1}{2}z^2\right] \end{pmatrix}\right]$$

The variance ($\mathrm{Var}_x$) and covariance ($\mathrm{Cov}_x$) are calculated from the untruncated normal $z \sim \mathcal{N}(w^Tx, \sigma^2)$. For all $x$, we have

$$\begin{pmatrix} v \\ \lambda \end{pmatrix}^T \begin{pmatrix} \mathrm{Var}_x[z]xx^T & -\mathrm{Cov}_x\left[\frac{1}{2}z^2, z\right]x \\ -\mathrm{Cov}_x\left[\frac{1}{2}z^2, z\right]x^T & \mathrm{Var}_x\left[\frac{1}{2}z^2\right] \end{pmatrix} \begin{pmatrix} v \\ \lambda \end{pmatrix}$$

$$=\sigma^2(x^Tv)^2 - 2\sigma^2 w^T x x^T v\lambda + \frac{1}{2}\sigma^2(2(w^Tx)^2 + \sigma^2)\lambda^2 := I(x)$$

Denote $\bar{X} = \mathbb{E}_{x\sim\mathcal{P}}\left[\sum xx^T\right]$. By assumption 2.4, we have $\bar{X} \succeq bI$, and thus we have

$$\mathbb{E}_{\mathcal{P}}[I(x)] = \mathbb{E}_{\mathcal{P}}[\sigma^2(x^Tv)^2 - 2\sigma^2 w^T x x^T v\lambda \frac{1}{2}\sigma^2(2(w^Tx)^2 + \sigma^2)\lambda^2]$$

$$=\sigma^2 v^T X v - 2\sigma^2 w^T X v\lambda + \frac{1}{2}\sigma^2(2w^T X w + \sigma^2)\lambda^2$$

$$=\sigma^2(x - w\lambda)^T X(x - w\lambda) + \frac{1}{2}\sigma^4\lambda^2$$

$$\geq b\sigma^2\|x - w\lambda\|^2 + \frac{1}{2}\sigma^4\lambda^2$$

So, it is larger than

$$\frac{b}{2b(1 + \|w\|^2/\sigma^2) + 1}(\sigma^2\|v\|^2 + \sigma^4\lambda^2)$$

Now we consider each summand again for the untruncated sum: we can write it as an integral in 2 variables with degree at most 4, as:

$$I(x) = \begin{pmatrix} v \\ \lambda \end{pmatrix}^T \begin{pmatrix} \mathrm{Var}_x[z]xx^T & -\mathrm{Cov}_x\left[\frac{1}{2}z^2, z\right]x \\ -\mathrm{Cov}_x\left[\frac{1}{2}z^2, z\right]x^T & \mathrm{Var}_x\left[\frac{1}{2}z^2\right] \end{pmatrix} \begin{pmatrix} v \\ \lambda \end{pmatrix}$$

$$= \iint_{\mathbb{R}\times\mathbb{R}} \frac{1}{2}(v^Tx)^2(y - z)^2 - \frac{1}{2}\lambda v^T x(y - z)^2(y + z) + \frac{1}{8}\lambda^2(y - z)^2(y + z)^2 d\mathbb{P}_x$$

Where $\mathbb{P}_x$ is the untruncated joint normal distribution $\mathcal{N}(w^Tx, \sigma^2)^{\otimes 2}$. This can easily be verified as a log-concave measure. Next, we define polynomial $p(y, z)$ as

$$p(y, z) = \frac{1}{2}(v^Tx)^2(y - z)^2 - \frac{1}{2}\lambda v^T x(y - z)^2(y + z) + \frac{1}{8}\lambda^2(y - z)^2(y + z)^2$$

Let $\alpha = I(x)\alpha(w, \sigma, x)^8/2^{12}\mathcal{C}^4$ where $\mathcal{C}$ is the constant defined in Theorem B.2. Since joint normal is a log-concave distribution, using Theorem B.2, plug in $d = q = 4$ and $\alpha$ we get

$$I(x)^{1/4}\alpha^{-1/4}\mathbb{P}\{|p(y, z)| \leq \alpha\} \leq 4\mathcal{C}$$

Hence we have $\mathbb{P}\{|p(y, z)| \leq \alpha\} \leq \frac{\alpha(w, \sigma, x)^2}{2}$. Notice that the polynomial $p$ is non-negative, therefore, we have a probability of $\mathbb{P}\{p(y, z) > \alpha\} > 1 - \frac{\alpha(w, \sigma, x)^2}{2}$. Since $\mathbb{P}\{p(y, z) \in S \times S\} =$

$\frac{\alpha(w,\sigma,x)^2}{2}$, we have $\mathbb{P}\left\{p(y,z) > \alpha | (y,z) \in S \times S\right\} > \frac{1}{2}$. So we can estimate

$$\begin{pmatrix} v \\ \lambda \end{pmatrix}^T \begin{pmatrix} \text{Var}'_x[z]xx^T & -\text{Cov}'_x\left[\frac{1}{2}z^2, z\right]x \\ -\text{Cov}'_x\left[\frac{1}{2}z^2, z\right]x^T & \text{Var}'_x\left[\frac{1}{2}z^2\right] \end{pmatrix} \begin{pmatrix} v \\ \lambda \end{pmatrix}$$

$$= \frac{1}{\alpha(w,\sigma,x)^2}\mathbb{E}\left[p(y,z)\right] \geq \frac{1}{\alpha(w,\sigma,x)^2}\frac{\alpha(w,\sigma,x)^2}{2}\alpha = \frac{\alpha(w,\sigma,x)^8}{2^{13}C^4}I^{(i)}$$

Here, the variance ($\text{Var}'_x$) and covariance ($\text{Cov}'_x$) are calculated from the truncated normal distribution $z \sim \mathcal{N}(w^Tx, \sigma^2, S)$.

Now, by taking both the minimum survival probability, $\alpha(w,\sigma,x)$ and the sum of the inequality above, we show that the lower bound of the Hessian Matrix is:

$$r\begin{pmatrix} \sigma^2 I & 0 \\ 0 & \sigma^4 \end{pmatrix} := \frac{\min_i \alpha(w,\sigma,x)^8}{2^{13}C^4}\frac{b}{2b(1 + \|w\|^2/\sigma^2) + 1}\begin{pmatrix} \sigma^2 I & 0 \\ 0 & \sigma^4 \end{pmatrix}$$

From the lower bound in Lemma B.3, we have

$$r \geq \exp\left(-\text{poly}(1/a)(1 + \frac{\beta}{\sigma_0^2})\right)\frac{b}{2b(1 + \|w\|^2/\sigma^2) + 1}$$

Since $\sigma > \text{poly}(1/a)\sigma_0$ and $\|w\|^2 < \beta$, we have

$$\mathbf{H} \succeq b\exp\left(-\text{poly}(1/a)(1 + \frac{\beta}{\sigma_0^2})\right)\begin{pmatrix} \sigma_0^2 I & 0 \\ 0 & \sigma_0^4 \end{pmatrix}$$

Using the same argument, we can derive Lemma 4.4, calculating the lower bound for the strong convexity of the minimum point. We can put $\sigma = \sigma^*$, $w = w^*$ and $\min_i \alpha(w,\sigma,x) \geq a$ due to the assumptions.

$$\mathbf{H}(v^*, \lambda^*) \succeq b\frac{a^8}{2^{13}C^4}\frac{1}{3 + 2b\|w^*\|^2/\sigma^{*2}}\begin{pmatrix} \sigma^{*2} I & 0 \\ 0 & \sigma^{*4} \end{pmatrix}$$

*Remark*: By Lemma B.3, we have $\alpha(w,\sigma,x) \geq \exp\left(-2s^2\left(-\log a + 2 + \frac{2\beta}{\sigma^{*2}}\right)\right)$. Notice that $\sigma^2 \geq \frac{1}{18(5 - 2\log a)}\sigma_0^2$, we have

$$\frac{b}{2b(1 + \|w\|^2/\sigma^2) + 1} \geq \frac{b}{2b(1 + \beta/\sigma^2) + 1} = \Omega\left(\frac{b}{b(1 + \beta/\sigma^2) + 1}\right)$$

Therefore, we can yield a bound of

$$\mathbf{H} \succeq \Omega\left(\exp\left(-16s^2\left(-\log a + 2 + \frac{2\beta}{\sigma^{*2}}\right)\right)\frac{b}{b(1 + \beta/\sigma^2) + 1}\right)\begin{pmatrix} \sigma^2 I & 0 \\ 0 & \sigma^4 \end{pmatrix}$$

Here, we can ignore the constant factor, since it is unknown what the $C$ is in Carbery & Wright (2001a). Also, we did not transform a general $\sigma$ in the projection set to $\sigma_0$ on purpose.

### B.3.2 Bounded Step Variance

Now, let's focus on the upper bound of the squared variance. To eliminate confusion let $y$ be the value of the dependent variable for which we are computing the gradient.

Let the $y \sim \mathcal{N}(w^{*T}x, \sigma^{2*})$ and $z \sim \mathcal{N}(w^Tx, \sigma^2)$. So, we have

$$\mathbb{E}_{\mathcal{P}}\left[\mathbb{E}\left[\|(y-z)x\|^2\right]\right] \leq 4 \cdot \mathbb{E}_{\mathcal{P}}\left[\mathbb{E}[\|(y - w^{*T}x)x\|^2 + \|(z - w^Tx)x\|^2]\right] + 4 \cdot \|x^T(w - w^*)x\|^2$$

For the first term, we have that by Lemma B.1, it holds that

$$4\mathbb{E}_{\mathcal{P}}\left[\mathbb{E}\left[\|(y - w^{*T}x)x\|^2\right]\right] \leq 4\mathbb{E}_{\mathcal{P}}\left[\mathbb{E}\left[(y - w^{*T}x)^2\right]\|x\|^2\right]$$

$$\leq 4(\sigma^*)^2(2 - 4\log(\alpha(w^*, x, \sigma^*)))\|x\|^2 \leq 4(2 - 4\log a)(\sigma^*)^2 \leq 4(2 - 4\log a)(s\sigma_0)^2$$

Similarly, we have

$$4\mathbb{E}_{\mathcal{P}}\left[\|(z - w^T x)x\|^2\right] \le 4\mathbb{E}_{\mathcal{P}}\left[(z - w^T x)^2\right]\|x\|^2$$

$$\le 4\sigma^2 \underset{\mathcal{P}}{\mathbb{E}}\left[2 - 4\log(\alpha(w, x, \sigma))\|x\|^2\right]$$

$$= 8\sigma^2 + 16\mathbb{E}_{\mathcal{P}}\left[-\sigma^2\log(\alpha(w, x, \sigma))\|x\|^2\right] \le 8\sigma^2 + 16\mathbb{E}_{\mathcal{P}}\left[-\sigma^2\log(\alpha(w, x, \sigma))\right]$$

By Lemma 4.5, we have

$$-\sigma^2\log(\alpha(w, x, \sigma))$$

$$\le \max\left(\sigma^2, 2\sigma^{*2}\right)\log\frac{1}{\alpha(w^*, \sigma^*, x)} + \max\left(0, 4\sigma^{*2} - 2\sigma^2\right) + 2\sigma^{*2}\left((w^T x - w^{*T}x)^2\right)$$

$$\le \sigma^{*2}(-s^2\log a + 4 + 2\beta)$$

Hence, we have

$$4\mathbb{E}_{\mathcal{P}}\mathbb{E}\left[\|(z - w^T x)x\|^2\right] \le 8\sigma^2 + 16\mathbb{E}_{\mathcal{P}}\left[-\sigma^2\log(\alpha(w, x, \sigma))\|x\|^2\right]$$

$$\le 8s^2\sigma_0{}^2 + 16s^2\sigma_0{}^2(4 + 2\beta - s^2\log a)$$

And for the last term, we have

$$4 \cdot \|x^T(w - w^*)x\|^2 \le 16\beta^2$$

because $\|w\|^2, \|w^*\|^2 \le \beta^2$ and $\|x\| \le 1$.

Now, let's deal with the squared gradient of $\lambda$. Notice that for $y \sim \mathcal{N}(w^{*T}x, \sigma^{2*}, S)$, we have:

$$\mathbb{E}[y^4] = (\mathbb{E}[y^2])^2 + \mathrm{Var}(y^2)$$

$$= 2(\mathbb{E}[(y - w^{*T}x)^2] + (w^{*T}x)^2)^2 + \mathrm{Var}(y^2)$$

$$\le 2((2 - 4\log\alpha(w^*, \sigma^*, x))\sigma^{*2} + (w^{*T}x)^2)^2 + 2(w^{*T}x)^2\sigma^{*2}O(\log\alpha(w^*, \sigma^*, x))$$

$$+ \sigma^{*4}O(\log\alpha(w^*, \sigma^*, x)^2)$$

$$\le 2(O(\log a)\mathrm{poly}(1/a)\sigma_0^2 + \beta)^2 + O(\log a)\mathrm{poly}(1/a)\beta\sigma_0^2 + \sigma_0^4 O(\log a)\mathrm{poly}(1/a)$$

$$= \mathrm{poly}(1/a)(\sigma_0{}^4 + \sigma_0^2\beta + \beta^2) \le \mathrm{poly}(1/a)(\sigma_0{}^4 + \beta^2)$$

The first inequality is due to 4.5 and B.4. The second inequality comes from the following three facts: because of the projection set, we have $\sigma^* = \mathrm{poly}(1/a)\sigma_0$, $(w^{*T}x)^2 \le \|w\|_2^2\|x\|_2^2 \le \beta$ by 2.2, and $\alpha(w^*, \sigma^*, x) \ge a$ by our assumption 2.3. The third inequality combines the term and by AM-GM inequality: $\beta\sigma_0 \le (\beta^2 + \sigma_0^4)/2$.

Now we are dealing with $z \sim \mathcal{N}(w^T x, \sigma^2, S)$, which is more complicated, but still has the same order bound.

$$\mathbb{E}[z^4] = (\mathbb{E}[z^2])^2 + \mathrm{Var}(z^2)$$

$$= 2(\mathbb{E}[(z - w^T x)^2] + (w^T x)^2)^2 + \mathrm{Var}(z^2)$$

$$\le 2((2 - 4\log\alpha(w, \sigma, x))\sigma^2 + (w^T x)^2)^2 + 2(w^T x)^2\sigma^2 O(\log\alpha(w, \sigma, x))$$

$$+ \sigma^4 O(\log\alpha(w, \sigma, x)^2)$$

The inequalities hold for the same reason as bounding $\mathbb{E}[y^4]$. By B.3 We know that $-\log\alpha(w, \sigma, x^{(i)}) \le \mathrm{poly}(1/a)(1 + \frac{\beta}{\sigma_0^2})$. Since we are estimating the parameters inside the projection set, we can derive $\sigma = \mathrm{poly}(1/a)\sigma_0$ and by 2.3, we have $(w^T x)^2 \le \|w\|_2^2\|x\|_2^2 \le \beta$, giving us

$$2((2 - 4\log\alpha(w, \sigma, x))\sigma^2 + (w^T x)^2)^2$$

$$\le 2((2 + \mathrm{poly}(1/a)(1 + \frac{\beta}{\sigma_0^2}))^2\sigma^2) + (w^T x)^2)$$

$$= 2((2 + \mathrm{poly}(1/a)(1 + \frac{\beta}{\sigma_0^2}))^2\sigma_0^2)\mathrm{poly}(1/a) + \beta)$$

$$= \mathrm{poly}(1/a)(\sigma_0^2 + \beta)^2 = \mathrm{poly}(1/a)(\sigma_0^4 + \beta^2)$$

The last inequality is due to the AM-GM inequality, since $\beta\sigma_0^2 \leq 1/2(\beta^2 + \sigma_0^4)$ and

$$2(w^T x)^2\sigma^2 O(\log\alpha(w,\sigma,x)) + \sigma^4 O(\log\alpha(w,\sigma,x)^2)$$

$$\leq 2\beta\sigma_0^2 O(\mathrm{poly}(1/a)(1 + \frac{\beta}{\sigma_0^2})) + \mathrm{poly}(1/a)\sigma_0^4(\mathrm{poly}(1/a)(1 + \frac{\beta}{\sigma_0^2}))^2$$

$$=2\beta O(\mathrm{poly}(1/a)(\beta + \sigma_0^2)) + \mathrm{poly}(1/a)((\sigma_0^2 + \beta))^2$$

$$=\mathrm{poly}(1/a)(\beta^2 + \beta\sigma_0^2 + \sigma^4) = \mathrm{poly}(1/a)(\sigma_0^4 + \beta^2)$$

Combining these, we get the result:

$$\mathbb{E}_{\mathcal{P}}\mathbb{E}\left[y^2/2 - z^2/2\right]^2 \leq \frac{1}{4}\mathbb{E}_{\mathcal{P}}\left[\mathbb{E}(y^4) + \mathbb{E}(z^4)\right]$$

$$\leq \mathbb{E}_{\mathcal{P}}\left[\mathrm{poly}(1/a)(\sigma_0^4 + \beta^2)\right] = \mathrm{poly}(1/a)(\sigma_0^4 + \beta^2) = \mathrm{poly}(1/a)(1 + \sigma_0^4 + \beta^2)$$

The gradient of $v$ is also within this bound. Thus, the bounded step variance is proved.

*Remark:* From the proof we can say

$$\mathbb{E}[y^4] = O(1 - \log a)(\sigma^{*4} + \beta^2)$$

And similarly, we have the bound for $\mathbb{E}(z^2)$ using Lemma B.3

$$\mathbb{E}[z^4] \leq 2((2 - 4\log\alpha(w,\sigma,x))\sigma^2 + (w^T x)^2)^2 + 2(w^T x)^2\sigma^2 O(\log\alpha(w,\sigma,x))$$
$$+\sigma^4 O(\log\alpha(w,\sigma,x)^2)$$

$$=2((2 + 8s^2(-\log a + 2 + \frac{2\beta}{\sigma^{*2}}))\sigma^2 + (w^T x)^2)^2$$

$$+2(w^T x)^2\sigma^2 O(2s^2(-\log a + 2 + \frac{2\beta}{\sigma^{*2}})) + \sigma^4 O(4s^4(-\log a + 2 + \frac{2\beta}{\sigma^{*2}})^2)$$

Notice that because $\sigma^2/\sigma^{*2} \leq s^2 = O(\frac{1-\log a}{a^2})$, and also $(w^T x)^2 \leq \|w\|^2\|x\|^2 \leq \beta$, the highest "degree" containing $a$ is $a^{-8}(1 - \log a)^{10}$ ($a^{-4}(1 - \log a)^4$ comes from $s^4$, additional $(1 - \log a)^2$ from the coefficients, and additionally $a^{-4}(1 - \log a)^4$ comes from a hidden $s^4$ produced by $\sigma^4/\sigma^{*4}$). Also, we can write $(w^T x)^2\sigma^2 = O(\sigma^4 + (w^T x)^4)$. Finally, we can write

$$\mathbb{E}[z^4] \leq O\left(\frac{(1 - \log a)^{10}}{a^8}(\beta^2 + \sigma^{*4})\right)$$

The terms in $\mathbb{E}[\|(y - z)x\|^2]$ have lower degrees of both in $1/a$, $\beta$, and $\sigma^*$. So, we can only add one to $\beta^2 + \sigma^{*4}$ as in the proof above. Therefore, we can write the total gradient as a bound of

$$O\left(\frac{(1 - \log a)^{10}}{a^8}(1 + \beta^2 + \sigma^{*4})\right)$$

This bound is greater than the bounded domain.

## B.4 Proof of Lemma 4.2

We estimate the projection set with the following procedure:

(1) We use OLS on $m$ samples $(x^{(i)}, y^{(i)})$, and get a estimated weight $w_0$

(2) Then we take another $m$ samples $(\bar{x}^{(i)}, \bar{y}^{(i)})$ and calculate the mean $\sigma_0^2 = \frac{1}{m}\sum_{i=1}^m(\bar{y}^{(i)} - w_0^T\bar{x}^{(i)})^2$.

First, we try to figure out the lower bound of $\sigma_0^2$. We prove a claim: $m$ is large (at least $m \geq 3$), for any $w$, there are at least $n/6$ (with probability $> 15/16$) samples such that $|\bar{y}^{(i)} - w^T\bar{x}^{(i)}| > \frac{a}{4}\sigma^*$. Notice that $\sigma_0^2 = \frac{1}{m}\sum|\bar{y}^{(i)} - w^T\bar{x}^{(i)}|^2$ for some empirical $w = w_0$. To finish the proof, we scale the whole distribution by multiplying by $\frac{1}{\sigma^*}$. Since we proved this, we have $\sigma_0^2 = \frac{1}{m}(\frac{n}{6} \times \frac{a^2}{16}\sigma^{*2}) \geq \frac{a^2}{96}\sigma^{*2}$. Now we can show that $\sigma^* = 1$ WLOG. In the proof of Lemma 9 in the paper Daskalakis et al. (2019),

we notice that in the worst case, $y_i - w^{*T} x^{(i)}$ are part of the distribution $D(a)$ (where $D(a)$ dominates $\text{Unif}(a/2, a/2) = U(a)$). for all $t > 0$ and $D \sim D(a)$ and $U \sim U(a)$, $\mathbb{P}(|D| < t) \leq \mathbb{P}(|U| < t)$ holds. Here, this means that even for the densest $[-\frac{a}{4}, \frac{a}{4}]$ it only take $1/2$. So, any $a/2$ window of $y$ takes at most $1/2$ of the probability. So we have

$$\mathbb{P}\left(|y^{(i)} - w^{*T} x^{(i)}| > \frac{a}{4}\right) > 1/2$$

Now, all we have left to prove is that if $X, X_i \sim \text{Ber}(1/2)$ $\mathbb{P}$, then $\mathbb{P}(X_1 + \cdots + X_n < n/6) < 1/16$ if $n > 4$. Thus, we need to prove that $\binom{n}{0} + \cdots + \binom{n}{r} < 2^n/16$ where $r = \lfloor (n-1)/6 \rfloor$.

For $n \leq 18$ it is easy to see this result by checking one by one. Regardless, for $13 \leq n \leq 18$, we have $(r+1)\binom{n}{r} < 2^n/16$. For $n \geq 19$, we use induction to prove the stronger bound of $(r+1)\binom{n}{r} < 2^n/16$. Now, suppose that this stronger statement holds for $n$, and we want to prove the same bound for $n + 6$. We already know that $(r+1)\binom{n}{r} < 2^n/16$. So, if we want to prove $(r+2)\binom{n}{r+1} < 2^{n+6}/16$, we need to show that

$$\frac{r+2}{(r+1)^2} \frac{(n+6)\cdots(n+1)}{(n-r+5)\cdots(n-r+1)} < 64$$

Further, we know that

$$\frac{r+2}{r+1} \leq \frac{4}{3} < 2$$

$$\frac{n+1}{r+1} \leq \frac{n+1}{n/6} \leq 6 \times \frac{14}{13} < 7$$

$$\frac{(n+6)\cdots(n+2)}{(n-r+5)\cdots(n-r+1)}$$
$$\leq \frac{(n+6)(n+5)(n+4)(n+3)(n+2)}{(5n/6+5)(5n/6+4)(5n/6+3)(5n/6+2)(5n/6+1)}$$
$$\leq \frac{19 \times 18 \times 17 \times 16 \times 15}{15 \times 14 \times 13 \times 12 \times 11} < 4$$

So, we get

$$\frac{r+2}{(r+1)^2} \frac{(n+6)\cdots(n+1)}{(n-r+5)\cdots(n-r+1)} < 2 \times 7 \times 4 < 64$$

Therefore, we prove the claim. Now we turn to the upper bound on the $\sigma_0^2$. Denote the least square estimator of $m$ samples to be $w_1$. So we have

$$\mathbb{E}[\sigma_0^2] = \mathbb{E}\left[\frac{1}{m}\sum_i (y^{(i)} - w_0^T x^{(i)})^2\right] \leq \frac{1}{2}\mathbb{E}\left[\sum_i (y^{(i)} - w_1^T x^{(i)})^2 + (w_0^T x^{(i)} - w_1^T x^{(i)})^2\right]$$

Since $w_1$ is the OLS estimator, we have

$$\mathbb{E}\left[\frac{1}{m}\sum_i (y^{(i)} - w_1^T x^{(i)})^2\right] \leq \mathbb{E}\left[\frac{1}{m}\sum_i (y^{(i)} - w^{*T} x^{(i)})^2\right] \leq \frac{1}{m}\sum_i (4 - 2\log a)\sigma^{*2} = (4 - 2\log a)\sigma^{*2}$$

The first inequality is because of OLS, and the second inequality is due to Lemma 4.5. Now we need to bound $\frac{1}{2}\mathbb{E}[\frac{1}{m}\sum_i (w_0^T x^{(i)} - w_1^T x^{(i)})^2]$. Notice that

$$(w_0^T x^{(i)} - w_1^T x^{(i)})^2 = ((w_0^T - w_1^T) x^{(i)})^2 \leq \left\|w_0^T - w_1^T\right\|^2 \left\|x^{(i)}\right\|^2 \leq \left\|w_0^T - w_1^T\right\|^2$$

So, we need this value to have a small norm. Also since we are doing OLS on the same distribution of data $(x, y)$, we can prove that there is a concentration of samples around the empirical estimator.

Moreover, notice that the weight formula for linear regression is $\left(\sum_i x^{(i)} x^{(i)T}\right)^{-1} \sum_i x^{(i)} y^{(i)}$ (here we abuse the notation, we are talking about general $m$ samples.) Or, we do an average, yielding

$(\frac{1}{m}\sum_i x^{(i)}x^{(i)T})^{-1}\frac{1}{m}\sum_i x^{(i)}y^{(i)}$. Therefore, if we have infinite samples, we get a final estimator $w_\infty = (\mathbb{E}(xx^T))^{-1}\mathbb{E}(yx)$ for $(x,y) \sim D^*$, where $D^*$ is defined in Section B.1. We also have $w_0, w_1 \to w_\infty$ if $m \to \infty$. Now, we are going to show how will they converge.

First, we investigate $\frac{1}{m}\sum_i x^{(i)}x^{(i)T}$. We need a theorem in Tropp (2015).

**Theorem B.5.** *(Combination of Theorem 1.6.2 and Section 1.6.3 in Tropp (2015)) Let $x_1,...,x_n$ be i.i.d. random vectors with dimension $p$. Let $Y = \frac{1}{n}\sum x_i x_i^T$ and $A = \mathbb{E}[x_i x_i^T]$. Assume that each one is uniformly bounded $\|x_k\| \le B$ for each $k = 1, \cdots, n$. Introduce the sum $Z = Y - A$ Then $\mathbb{P}(\|Z\|_2 \ge t) \le 2p \cdot \exp\left(\frac{-nt^2/2}{B\|A\|_2+Bt/3}\right)$ for all $t \ge 0$. Here $\|A\|_2$ is the spectral norm.*

Let $X_0 = \mathbb{E}(xx^T)$. We can choose a big enough $m$ such that with probability $1 - \frac{1}{64}$, the spectral norm of $\frac{1}{m}\sum_i x^{(i)}x^{(i)T} - X_0$ does not exceed $\delta$; $\delta$ is another constant determined later. This $\delta$ is smaller than $b$ and $X_0 \succeq bI$ by the assumptions. So, $\frac{1}{m}\sum_i x^{(i)}x^{(i)T}$ is positive definite and we can take the inverse.

Now, let's calculate the inverse. Let $\bar{X} = \frac{1}{m}\sum_i x^{(i)}x^{(i)T}$. Notice that

$$\bar{X}^{-1} - X_0^{-1} = X_0^{-1}(X_0 - \bar{X})\bar{X}^{-1}$$

has spectral norm at most $\frac{\delta}{b(b-\delta)}$. Since $X_0$ has a spectral norm $\ge b$, $X_0 - \bar{X}$ has spectral norm $\le \delta$ therefore $\bar{X}$ has spectral norm $\ge b - \delta$.

Then we investigate $\bar{x} = \frac{1}{m}\sum_i y^{(i)}x^{(i)}$. Let $x_0 = \mathbb{E}[yx]$. We use Chebyshev's inequality in the vector form. Notice that

$$\mathbb{P}(\|\bar{x} - x_0\| \ge \delta') \le \frac{\mathbb{E}[\|\bar{x} - x_0\|^2]}{\delta'^2}$$

In our case, we know that all $y, x$ are independently chosen. So, we have

$$\mathbb{E}[\|\bar{x} - x_0\|^2] \le \mathbb{E}[\|\bar{x}\|^2] = \frac{1}{m}\mathbb{E}[\|y \cdot x\|^2]$$

$$= \frac{1}{m}\left(\mathbb{E}\left[\frac{1}{2}\left(\left\|(y - w^{*T}x)^2 \cdot x\right\|^2 + \left\|(w^{*T}x)x\right\|^2\right)\right]\right) \le \frac{1}{2m}((4 - 2\log a)\sigma^{*2} + \beta)$$

Here, we use both Lemma 4.5 and assumption 1.1 that $\|x\| \le 1, \|w\|^2 \le \beta$. From this we can also derive that

$$\mathbb{E}[\|x_0\|] \le \sqrt{\mathbb{E}[\|x_0\|^2]} = \sqrt{(4 - 2\log a)\sigma^{*2} + \beta}$$

We also make $m$ large enough to have the probability of at most $1/64$, so that $\|\bar{x} - x_0\| \ge \delta'$.

Notice that we have the estimator $w = \bar{X}^{-1}\bar{x}_0$. This gives us

$$\left\|\bar{X}^{-1}\bar{x} - X_0^{-1}x_0\right\| = \left\|\bar{X}^{-1}(\bar{x} - x_0) + (\bar{X}^{-1} - X_0^{-1})x_0\right\|$$

$$\le \left\|\bar{X}^{-1}(\bar{x} - x_0)\right\| + \left\|(\bar{X}^{-1} - X_0^{-1})x_0\right\| \le \frac{1}{b-\delta}\delta' + \frac{\delta}{b(b-\delta)}\sqrt{(4 - 2\log a)\sigma^{*2} + \beta}$$

(Notice for any vector $v$ and square matrix $A$ if $\|v\| \le a, \|A\|_2 \le b$, then $\|Av\| \le ab$. This is because the $L_2$ norm for a vector equals to its spectral norm, and we have $\|AB\|_2 \le \|A\|_2\|B\|_2$) .

Since there is a $1/64$ probability that $\|\bar{x} - x_0\| \le \delta'$ does not hold, and there is also a $1/64$ probability that $\left\|\bar{X} - X_0\right\|_2 \le \delta'$ does not hold, this $\left\|\bar{X}^{-1}\bar{x} - X_0^{-1}x_0\right\| \le \frac{1}{b-\delta}\delta' + \frac{\delta}{b(b-\delta)}\sqrt{(4 - 2\log a)\sigma^{*2} + \beta}$ holds with a probability of at least $1 - 1/32$. That means, $w_0$ and $w_1$ each have a $1 - 1/32$ probability of being close to the convergence limit, and a probability of at most $1 - 1/16$ that both of them are close.

Take $\delta = \frac{\sigma^* b^2}{4\sqrt{(4-2\log a)\sigma^{*2}+\beta}}$ and $\delta' = b\sigma^*/4$. So, we have at most $1 - 1/16$ probability that both $w_0$ and $w_1$ are

$$\frac{1}{b-\delta}\delta' + \frac{\delta}{b(b-\delta)}\sqrt{(4 - 2\log a)\sigma^{*2} + \beta} \le \frac{\sigma^*}{2}$$

close to $w_\infty$. Therefore, the norm between $w_0$ and $w_1$ is at most $\sigma^*$.

To achieve $\delta$ difference of $\bar{X} - X_0$, we may need $2p \cdot \exp\left(\frac{-nt^2/2}{B\|A\|_2 + Bt/3}\right) \leq 1/64$. In our case, we have $p = k, t = \delta, B = 1$ and $\|A\|_2$ also has a upper bound 1. So, we need at least $\log(128k)\frac{1+\delta/3}{\delta^2} \leq \log(128k)\frac{32((4-2\log a)\sigma^{*2}+\beta)}{b^4\sigma^{*2}}$ samples.

Also, to achieve $\delta'$ difference of $\bar{x} - x_0$, we may need $\frac{1}{2m}((4 - 2\log a)\sigma^{*2} + \beta)\frac{1}{(b\sigma^*/4)^2} \leq 1/64$. So, we need at least $\frac{512((4-2\log a)\sigma^{*2}+\beta)}{b^2\sigma^{*2}}$ samples.

These two bounds of samples are covered with "poly" constraints of the informal theorem 1.4 and the first inequality of theorem 3.2.

Overall, we have

$$\mathbb{E}[\sigma_0^2] = \mathbb{E}\left[\frac{1}{m}\sum_i(y^{(i)} - w_0^T x^{(i)})^2\right] \leq \frac{1}{2}\mathbb{E}\left[\sum_i(y^{(i)} - w_1^T x^{(i)})^2 + (w_0^T x^{(i)} - w_1^T x^{(i)})^2\right]$$

$$\leq \frac{1}{2}((4 - 2\log a)\sigma^{*2} + \|w_0^T - w_1^T\|^2) \leq \frac{1}{2}((4 - 2\log a)\sigma^{*2} + \sigma^{*2}) = \frac{1}{2}((5 - 2\log a)\sigma^{*2})$$

By Markov's Inequality, we have a probability of $1/16$ that $\sigma_0^2 \leq 16(\mathbb{E}(\sigma_0^2))$. Now, we have $1/8$ probability ($1/16$ for Markov, $1/16$ for the concentration) to hold $\sigma_0^2 \leq 16(\mathbb{E}(\sigma_0)) \leq 8(5 - 2\log a)\sigma^{*2}$. Therefore, for the whole projection set, it takes $\geq 1 - 1/8 - 1/16 = 13/16$ probability to hold.

## B.5 Proof of the Smoothness

The smoothness can be given by this bound:

**Theorem B.6.** *The Hessian of $\bar{\ell}(v, \lambda)$ satisfies $\mathbf{H}^2(v, \lambda) \preceq \gamma^2 I$ where $\gamma = \text{poly}(1/a, \frac{1}{\sigma_0}, \sigma_0, \beta)$. Hence $\bar{\ell}$ is $\gamma$ smooth.*

Denote that $s = \frac{\max_{\sigma \in D_r} \sigma}{\min_{\sigma \in D_r} \sigma} = \sqrt{8(5 - 2\log a)\frac{96}{a^2}}$ is the largest possible difference between the variances. Since we already have $\mathbf{H}$ is symmetric and positive definite, $\mathbf{H}^2$ is symmetric and positive definite. So, by the Cauchy-Schwarz inequality, we have

$$\mathbf{H}^2 = \left(\mathbb{E}_{x\sim\mathcal{P}}\left[\mathbf{H}_x\right]\right)^2 \preceq \mathbb{E}_{x\sim\mathcal{P}}\left[\mathbf{H}_x^2\right]$$

Since for each sample $(x^{(i)})$ we can calculate

$$\mathbf{H}_x^2 = \begin{pmatrix} Ax_x^T & -Bx \\ -Bx^T & C \end{pmatrix}^2$$

$$= \begin{pmatrix} (A^2 + B^2\|x\|^2)xx^T & -(A+C)B\|x\|^2 x \\ -(A+C)B\|x\|^2 x^{(i)T} & (B\|x\|^2)^2 + C^2 \end{pmatrix}$$

$$\preceq 2\begin{pmatrix} (A^2 + B^2\|x\|^2)xx^T & 0 \\ 0 & (B\|x\|^2)^2 + C^2 \end{pmatrix}$$

$$\preceq 2\begin{pmatrix} (A^2 + B^2)I & 0 \\ 0 & B^2 + C^2 \end{pmatrix}$$

$$\preceq 2\begin{pmatrix} (A^2 + AC)I & 0 \\ 0 & AC + C^2 \end{pmatrix}$$

where

$$A = \text{Var}_D[z], B = \text{Cov}_D\left[\frac{1}{2}z^2, z\right], C = \text{Var}_D\left[\frac{1}{2}z^2\right]$$

and the distribution $D = \mathcal{N}(w^T x, \sigma^2, S)$,

we can deduce the first inequality this way. Since the matrix $\mathbf{H}_x^2$ is positive definite, it is also positive semi-definite. Further, if the matrix $\begin{pmatrix} A & B \\ B^T & C \end{pmatrix}$ is positive semi-definite, the matrix $\begin{pmatrix} A & -B \\ -B^T & C \end{pmatrix}$ is also positive- semi-definite. Thus, we conclude

$$\begin{pmatrix} A & B \\ B^T & C \end{pmatrix} \preceq 2 \begin{pmatrix} A & 0 \\ 0 & C \end{pmatrix}$$

The second inequality comes from $B^2 > 0$ and $d = \|x\|_2^2 < 1$. Finally, the third inequality comes from $B^2 < AC$.

Thus, we can write

$$\gamma^2 = 2\mathbb{E}_{\mathcal{P}}\Big[A + C\Big]^2 = 2\mathbb{E}_{\mathcal{P}}\left[\text{Var}_{D(x)}([z] + \text{Var}_{D(x)}[\tfrac{1}{2}z^2)]\right]^2$$

where $D(x) = \mathcal{N}(w^T x, \sigma^2, S)$. Also, by Lemma B.4, we have

$$(A + C)^2 = \text{poly}\left(\log \frac{1}{\alpha(w, \sigma, x)}\sigma^2 + \log^2 \frac{1}{\alpha(w, \sigma, x)}\sigma^4 + \log \frac{1}{\alpha(w, \sigma, x)}(w^T x)^2 \sigma^2\right)$$

Since $\log \frac{1}{\alpha(w,\sigma,x)}$ can be written as $\text{poly}(1/a)(1 + \beta/\sigma_0^2)$ by Lemma B.3, $\sigma$ can be written as $\text{poly}(\sigma_0, 1/a)$, and $|w^T x| \leq \beta$. Then finally we can write

$$\gamma^2 = \text{poly}\left(1/a, \beta, \sigma_0 + \frac{1}{\sigma_0}\right)$$

Now, we will prove the smoothness. For any two vectors $\theta_1 = (v_1, \lambda_1)$ and $\theta_2 = (v_2, \lambda_2)$, we define the unit vector with same direction of $(v_2, \lambda_2) - (v_1, \lambda_1)$ as $\mathbf{u}$. Since the projection set written in $v, \lambda$ is convex, we claim that all of the points along the line segment $(v_2, \lambda_2)(v_1, \lambda_1)$ are within the projection set. Hence,

$$\left\|\nabla\bar{\ell}(v_2, \lambda_2) - \nabla\bar{\ell}(v_1, \lambda_1)\right\| = \left\|\int_0^{\|\theta_2 - \theta_1\|} \frac{\partial \nabla\bar{\ell}}{\partial \mathbf{u}}\bigg|_{(v_1,\lambda_1)+\mathbf{u}t} \mathbf{u}\,\mathrm{d}t\right\|$$

$$\leq \int_0^{\|\theta_2 - \theta_1\|} \left\|\frac{\partial \nabla\bar{\ell}}{\partial \mathbf{u}}\bigg|_{(v_1,\lambda_1)+\mathbf{u}t} \mathbf{u}\right\|\mathrm{d}t = \int_0^{\|\theta_2 - \theta_1\|} \sqrt{\mathbf{u}^T \mathbf{H}((v_1, \lambda_1) + \mathbf{u}t)^T \mathbf{H}((v_1, \lambda_1) + \mathbf{u}t)\mathbf{u}}\,\mathrm{d}t$$

$$\leq \int_0^{\|\theta_2 - \theta_1\|} \sqrt{\mathbf{u}^T \gamma^2 I \mathbf{u}}\,\mathrm{d}t = \gamma\|\theta_2 - \theta_1\|$$

Thus, we derive that $L_{\mathcal{D}}$ is $\gamma$-smooth.

### B.6 Proof of Corollary 4.6

*Proof.* By Theorem 4.1, we can choose $\rho$ to be $\text{poly}(\beta, \sigma_0 + \frac{1}{\sigma_0}, \frac{1}{a})$, and $\zeta = \exp\left(-\text{poly}(1/a)(1 + \frac{\beta}{\sigma_0^2})\right)\min\{\sigma_0^2, \sigma_0^4\}$. Then using Section 4.4 the assumptions of Theorem 3.5 are satisfied and applying Theorem 3.5 the Lemma follows. □

### B.7 Detailed proof of Theorem 3.2

The notation is the same as in Section 4.5, and we present the details of the two cases considered in that section. Here, the estimation below does not contain the number of samples for $n$ to to define projection $D$. In Section 4.2, we need the number of samples as $O\left(\frac{(1-\log a)\sigma^{*2}+\beta}{b^4\sigma^{*2}}\right)$. This is bounded above by the right hand side of 3.2: $n$ must be at the order of $\text{poly}(\frac{\sigma^* \cdot \beta}{a \cdot b}, \frac{1}{a \cdot \sigma^*})$ and the order of $\text{poly}(\sigma^*, \frac{1}{b}) \cdot \exp\left(\text{poly}(\frac{\beta}{a \cdot \sigma^*})\right)$.

**Case** $\eta_t = c/\sqrt{t}$. First, combining Theorem 3.4 and Theorem 4.1 and the projection set, we get the corollary 4.3. We can derive the corollary as follows: by Theorem 4.1 and the rest of Section 4.3

,we know that the step variance and the domain is bounded by some polynomial of $\sigma_0, \beta, \frac{1}{a}, 1/\sigma_0$. By Theorem 3.4, if the final $(w^*, \sigma^{*2}) \in D$, we have, after $n$ steps, $\mathbb{E}\left[\bar{\ell}(\hat{v}, \hat{\lambda}) - \bar{\ell}(v^*, \lambda^*)\right] \leq \frac{\text{poly}(\frac{\sigma_0 \cdot \beta}{a \cdot b}, \frac{1}{a \times \sigma_0}) \cdot \log(n)}{n^{1/2}}$. Notice that from the definition of the projection set, $\sigma_0/\sigma^*$ and $\sigma^*/\sigma_0$ both are bounded by $\text{poly}(1/a)$. Therefore, we can rewrite this bound to $\mathbb{E}\left[\bar{\ell}(\hat{v}, \hat{\lambda}) - \bar{\ell}(v^*, \lambda^*)\right] \leq \frac{\text{poly}(\frac{\sigma^* \cdot \beta}{a \cdot b}, \frac{1}{a \times \sigma^*}) \cdot \log(n)}{n^{1/2}}$.

Our next step is to transform the optimality in function values to closeness to the true parameters. To do so we use the strong convexity of $\bar{\ell}$ at the optimum.

By Lemma 4.5, we have:

$$\bar{\ell}(v^*, \lambda^*) = \mathbb{E}_{x \in \mathcal{P}} \mathbb{E}_{y} \left[\frac{(y - w^{*T}x)^2}{2\sigma^{*2}} + \log \alpha(w^*, \sigma^*, x)\right] \leq 4 - 2\log \min_{x \in \mathcal{P}} \alpha(w^*, \sigma^*, x) \leq 4 - 2\log a$$

We receive the first inequality by ignoring the $\log \alpha(w^*, \sigma^*, x)$ term and the fact that the expectation is smaller then the maximum. By applying Markov's inequality we get

$$\mathbb{P}\left(\bar{\ell}(\hat{v}, \hat{\lambda}) - \bar{\ell}(v^*, \lambda^*) > 5\mathbb{E}\left[\bar{\ell}(\hat{v}, \hat{\lambda}) - \bar{\ell}(v^*, \lambda^*)\right]\right) < \frac{1}{7}. \tag{B.1}$$

The inequality implies that, with a probability of at least $6/7$, the actual difference is at most $7$ times its expectation. The constant $7$ can be absorbed by the poly term. Also, the probability that $(v^*, \lambda^*) \in D$ is at least $13/16$ by Lemma 4.2. Combining these two, we have at least $13/16 - 1/7 > 2/3$ probability to hold:

$$\bar{\ell}(\hat{v}, \hat{\lambda}) - \bar{\ell}(v^*, \lambda^*) \leq \mathbb{E}\left[\bar{\ell}(\hat{v}, \hat{\lambda}) - \bar{\ell}(v^*, \lambda^*)\right] \leq \frac{\text{poly}(\frac{\sigma^* \cdot \beta}{ab}, \frac{1}{a \times \sigma^*}) \cdot \log(n)}{n^{1/2}} \tag{B.2}$$

Now by applying Lemma 4.4, we transform the difference in functions to the $L_2$ norm distance. Since $\hat{v}, \hat{\lambda}$ converge to $v^*, \lambda^*$, when they have small distance, we can approximate the rate using the convexity of the minimum. Therefore, we have

$$\|\hat{v} - v^*\|^2 + (\hat{\lambda} - \lambda^*)^2 \leq \frac{2}{\delta}\left(\bar{\ell}(\hat{v}, \hat{\lambda}) - \bar{\ell}(v^*, \lambda^*)\right) \leq \frac{\text{poly}(\frac{\sigma^* \cdot \beta}{ab}, \frac{1}{a \times \sigma^*}) \cdot \log(n)}{\delta \cdot n^{1/2}} \tag{B.3}$$

where

$$\mathbf{H}(v^*, \lambda^*) \succeq b\frac{a^8}{2^{13}\mathcal{C}^4}\frac{1}{3 + 2b\|w^*\|^2/\sigma^{*2}}\begin{pmatrix} \sigma^{*2}I & 0 \\ 0 & \sigma^{*4} \end{pmatrix} \succeq \delta I$$

Notice that $\|w\|^2 \leq \beta$. Therefore $\delta^{-1} = \text{poly}\left(\frac{\sigma^* \cdot \beta}{a \cdot b}, \frac{1}{a \cdot \sigma^*}\right)$, and this term can be absorbed into the term $\text{poly}(\frac{\sigma^* \cdot \beta}{a \times b}, \frac{1}{a\sigma^*})$.

Finally, by Cauchy-Schwartz Inequality, we have

$$\|\hat{w} - w^*\| + \left|\hat{\sigma}^2 - \sigma^{*2}\right| \leq \sqrt{2\left(\|\hat{w} - w^*\|^2 + \left|\hat{\sigma}^2 - \sigma^{*2}\right|^2\right)} \tag{B.4}$$

And for the $w$ part, we parameterize back and we get

$$\|\hat{w} - w^*\|^2 \leq \left\|\frac{\hat{v}}{\hat{\lambda}} - \frac{v^*}{\lambda^*}\right\|^2 \leq 2\left(\left\|\frac{\hat{v}}{\hat{\lambda}} - \frac{\hat{v}}{\lambda^*}\right\|^2 + \left\|\frac{\hat{v}}{\lambda^*} - \frac{v^*}{\lambda^*}\right\|^2\right)$$

$$\leq \frac{\|\hat{v}\|^2}{\lambda^{*2}\hat{\lambda}^2}(\hat{\lambda} - \lambda^*)^2 + \frac{\|\hat{v} - v^*\|^2}{\lambda^{*2}} \leq \text{poly}(1/a, \sigma^*, \beta)\left\|(\hat{v}, \hat{\lambda}) - (v^*, \lambda^*)\right\|^2 \tag{B.5}$$

The last inequality is because of the projection set, we have $\hat{\lambda}, \lambda^* = \text{poly}(1/a)\sigma^{*-2}$. Also, by Assumption 2.2 we have $\|\hat{v}\| = \|\hat{w}\|/\hat{\sigma}^2 \leq \sqrt{\beta}/(\text{poly}(1/a)\sigma^{*2})$. For the $\sigma$, we have

$$|\hat{\sigma}^2 - \sigma^{*2}|^2 = \left\|\frac{1}{\hat{\lambda}} - \frac{1}{\lambda^*}\right\|^2 \leq \frac{1}{\lambda^{*2}\hat{\lambda}^2}(\hat{\lambda} - \lambda^*)^2 \leq \text{poly}(1/a, \sigma^*)\left\|(\hat{v}, \hat{\lambda}) - (v^*, \lambda^*)\right\|^2 \tag{B.6}$$

The last inequality is also derived from the projection set. Combining inequality (B.1) with inequality (B.6), we have, with a probability of at least $2/3$,

$$\|\hat{w} - w^*\| + \left|\hat{\sigma}^2 - \sigma^{*2}\right| \leq \sqrt{2\left(\|\hat{w} - w^*\|^2 + \left|\hat{\sigma}^2 - \sigma^{*2}\right|^2\right)}$$

$$\leq \sqrt{2\mathrm{poly}(1/a, \sigma^*, \beta)\left\|(\hat{v}, \hat{\lambda}) - (v^*, \lambda^*)\right\|^2} \leq \sqrt{2\mathrm{poly}(1/a, \sigma^*, \beta)\frac{\mathrm{poly}(\frac{\sigma^* \cdot \beta}{ab}, \frac{1}{a \cdot \sigma^*}) \cdot \log(n)}{\delta \cdot n^{1/2}}}$$

$$= \frac{\mathrm{poly}(\frac{\sigma^* \cdot \beta}{ab}, \frac{1}{a \cdot \sigma^*}) \cdot \log(n)}{n^{1/4}}$$

And therefore, inequality (3.1) of Theorem 3.2 follows.

*Remark:* We can derive in the theorem 2 in Shamir & Zhang (2013) that the bound is $O(\rho^2 \frac{1 + \log n}{\sqrt{n}})$. We have already written the bound for $\mathbf{H}$ and step variance in the form of $\sigma^*$. Therefore, the right hand side of inequality B.2 is

$$O\left(\frac{(1 - \log a)^{10}}{a^8}(1 + \beta^2 + \sigma^{*4})\frac{\log n}{\sqrt{n}}\right)$$

And $\delta^{-1} = \frac{3 + 2b\beta/\sigma^{*2}}{ba^8 \min(\sigma^{*2}, \sigma^{*4})} = O\left(\frac{(\sigma^{*2} + \beta)(1 + \sigma^{*2})}{ba^8 \sigma^{*6}}\right)$. Here, we used a trick that $\min(x^2, x^4) = \Omega(\frac{x^4}{1 + x^2})$

By equations B.5 and B.6, the factor to be multiplied from $v, \lambda$ to $w$, is $\max(\frac{1}{\lambda^{*2}}, \frac{\|v\|^2}{\hat{\lambda}^2 \lambda^{*2}}) = \max(\sigma^{*4}, \sigma^{*4}\|\hat{w}\|^2) = O((\sigma^*)^4(1 + \beta))$ the factor to be multiplied from $\lambda$ to $\sigma$ is $\frac{1}{\hat{\lambda}^2 \lambda^{*2}} = \sigma^{*2}\hat{\sigma}^2 = O(\sigma^{*4}s^2) = O(\sigma^{*4}\frac{1 - \log a}{a^2})$. So, the overall multiplying factor is $O(\sigma^{*4}\frac{1 - \log a}{a^2}(1 + \beta))$

Therefore, by multiplying these bounds and square root, we have a bound of

$$O(\sqrt{\frac{(1 - \log a)^{10}}{a^8}(1 + \beta^2 + \sigma^{*4})\frac{\log n}{\sqrt{n}}\frac{(\sigma^{*2} + \beta)(1 + \sigma^{*2})}{ba^8 \sigma^{*6}}\sigma^{*4}\frac{1 - \log a}{a^2}(1 + \beta)})$$

Which can be simplify to

$$O\left(\frac{(1 - \log a)^{5.5}}{a^9}\frac{\sqrt{\log n}}{\sqrt[4]{n}}\frac{(1 + \beta^2)(\sigma^{*3} + \sigma^{*-1})}{\sqrt{b}}\right)$$

**Case $\eta_t = 1/(\zeta t)$.** We just plug in the result in Theorem 4.1 into Theorem 3.5 and in Section B.6 we have proved Corollary 4.6. That is, if $w^*, \sigma^* \in D$, we have

$$\mathbb{E}\left[\left\|(\hat{v}, \hat{\lambda}) - (v^*, \lambda^*)\right\|^2\right] \leq \mathrm{poly}\left(\beta, \sigma_0 + \frac{1}{\sigma_0}, \frac{1}{a}\right) \times \exp\left(\mathrm{poly}(1/a)\left(1 + \frac{\beta}{\sigma_0^2}\right)\right)\frac{1}{b^2 n}.$$

Applying Markov's inequality again as inequality (B.1), we have

$$\mathbb{P}\left(\left\|(\hat{v}, \hat{\lambda}) - (v^*, \lambda^*)\right\|^2 < 7\mathbb{E}\left[\left\|\hat{v}, \hat{\lambda} - (v^*, \lambda^*)\right\|^2\right]\right) < 1/7 \tag{B.7}$$

Similarly to the proof of inequality (3.1) above, of Theorem 3.2, we have that with probability at least $2/3$ it holds that

$$\left\|(\hat{v}, \hat{\lambda}) - (v^*, \lambda^*)\right\|^2 \leq \frac{\mathrm{poly}(\sigma_0) \cdot \exp\left(\mathrm{poly}(\frac{\beta}{a \cdot (\sigma_0)})\right)}{b^2 \cdot n}$$

Again, using the same inequality (B.4) to (B.6), we can transform the bound from square bound to a linear one:

$$\|\hat{w} - w^*\| + \left|\hat{\sigma}^2 - \sigma^{*2}\right| \leq \sqrt{2\text{poly}(1/a, \sigma^*, \beta) \left\|(\hat{v}, \hat{\lambda}) - (v^*, \lambda^*)\right\|^2}$$

$$\leq \sqrt{2\text{poly}(1/a, \sigma^*, \beta) \frac{\text{poly}(\sigma_0) \cdot \exp\left(\text{poly}(\frac{\beta}{a \cdot (\sigma_0)})\right)}{b^2 \cdot n}}$$

$$= \frac{\text{poly}(\sigma_0) \cdot \exp\left(\text{poly}(\frac{\beta}{a \cdot \sigma_0})\right)}{b\sqrt{n}}$$

We get the last inequality because the $\text{poly}(1/a, \sigma^*, \beta)$ can be absorbed into $\text{poly}(\sigma_0)$ and exponential term of $\text{poly}(\beta/\alpha\sigma_0)$. Therefore, the inequality (3.2) of Theorem 3.2 follows.

*Remark*: Again, we can calculate the bound by plugging Theorem 4.1 into 3.5. This gives us a difference of $\frac{4\rho^2}{\zeta^2 t}$, where $\rho$ is the bound on the step variance in Section B.3.2 and $\zeta$ is the lower bound in the B.3.1. Specifically, we have

$$\rho^2 = O\left(\frac{(1 - \log a)^{10}}{a^8}(1 + \beta^2 + \sigma^{*4})\right)$$

and from the remark in the B.3.1, we have

$$\zeta = \Omega\left(\exp\left(-16s^2\left(-\log a + 2 + \frac{2\beta}{\sigma^{*2}}\right)\right)\frac{b}{b(1 + \beta/\sigma^2) + 1}\right)\min(\sigma^2, \sigma^4)$$

for all the possible $\sigma$ in the projection set. Since we know that $\sigma^2/\sigma^{*2}$ we may say $\sigma^2/\sigma^{*2} \geq O\left(\frac{a^2}{1 - \log a}\right)$. Also $s = \frac{96 \times 8(5 - 2\log a)}{a^2}$. For $\frac{(b(1 + \beta/\sigma^2) + 1)}{b\min(\sigma^2, \sigma^4)}$, the numerator is $O(1 + \beta/\sigma^2) = O(1 + \frac{\beta/(1 - \log a)\sigma^{*2}}{a^2}) = O((1 + \beta/\sigma^{*2})\frac{1 - \log a}{a^2})$, and the denominator is $b\min(\sigma^2, \sigma^4) = \Omega\left(b\frac{\sigma^4}{1 + \sigma^2}\right) = \Omega\left(b\frac{\sigma^{*4}}{1 + \sigma^{*2}}\frac{(1 - \log a)^2}{a^4}\right)$. Therefore, we can write $\frac{1}{\zeta}$ as

$$O\left(\exp\left(\frac{9 \times 2^{14}(1 - \log a)}{a^2}\left(-\log a + 2 + \frac{2\beta}{\sigma^{*2}}\right)\right)\frac{1 + \beta/\sigma^{*2}}{b\sigma^{*4}}(1 + \sigma^{*2})\frac{(1 - \log a)^3}{a^6}\right)$$

Since we know that $\rho^2 = O(\frac{(1 - \log a)^{10}}{a^8}(1 + \beta^2 + \sigma^{*4}))$ and the coefficient $O(\sigma^{*4}\frac{1 - \log a}{a^2}(1 + \beta)) =: K$ for transforming $v, \lambda$ to $w, \sigma$, we can take the square root of this term, and calculate a bound for the final estimation. This is given by taking $\sqrt{\frac{4\rho^2}{\zeta^2 n}}K$ and then counting the both highest and lowest degrees of $\beta, \sigma^*, a$

$$O\left(\exp\left(\frac{3 \times 2^{12}(5 - 2\log a)}{a^2}\left(-\log a + 2 + \frac{2\beta}{\sigma^{*2}}\right)\right)\frac{(1 - \log a)^{8.5}}{a^{15}}(1 + \beta^{2.5})(\sigma^{*-1} + \sigma^{*3})\frac{1}{b\sqrt{n}}\right)$$

## B.8   Proof of Theorem 5.1 and Corollary 5.2

We cite a lemma in the multivariate version of Proposition 2 in Leluc & Portier (2020).

**Definition B.7.** A stochastic algorithm is a sequence $(x_k)_{k \geq 0}$ of random variables defined in a probability space $(\Omega, \mathcal{F}, P)$ and valued in $\mathbb{R}^d$. Define $(\mathcal{F}_k)_{k \geq 0}$ as the natural $\sigma$-field associated to the stochastic algorithm $(x_k)_{k \geq 0}$, i.e., $\mathcal{F}_k = \sigma(x_0, x_1, \cdots, x_k), k \geq 0$. A policy is a sequence of random probability measures $(P_k)_{k \geq 0}$, each defined on a measurable space $(S, \mathcal{S})$ that are adapted to $\mathcal{F}_k$.

**Definition B.8.** Given a policy $(P_k)_{k \geq 0}$ and a learning rates sequence $(\alpha_k)_{k \geq 1}$ of positive numbers, the SGD algorithm is defined by the update rule $x_k = x_{k-1} - \alpha_k C_k g(x_{k-1}, \xi_k)$ where $\xi_k \sim P_{k-1}$ with $g : \mathbb{R}^d \times S \to \mathbb{R}^d$

**Theorem B.9.** *Proposition 2 in Leluc & Portier (2020). If we suppose that the assumptions below are fulfilled:*

1. *The gradient estimation of $F$ is unbiased, that is, for $k \geq 1$, the expected sampled gradient is equal to the total gradient. ($\mathbb{E}(g(x_{k-1}, \xi_k)|\mathcal{F}_{k-1}) = \nabla F(x_{k-1})$)*

2. *The sequence $(\alpha_k)_{k \geq 1}$ is positive, decreasing to 0, and satisfies the Robbins-Monro condition: $\sum_{k \geq 1} \alpha_k = +\infty$ and $\sum_{k \geq 1} \alpha_k^2 < \infty$.*

3. *The objective function is $L-$smooth.*

4. *The objective function has only one minimum point $x^*$ and $\lim_{\|x\| \to \infty} F(x) = \infty$*

5. *With probability 1, there exist $0 \leq \mathcal{L}, \sigma^2 < \infty$ such that*

$$\forall x \in \mathbb{R}^d, \forall k \in \mathbb{N}, \mathbb{E}(||g(x_{k-1}, \xi_k)||^2|\mathcal{F}_{k-1}) \leq 2\mathcal{L}(F(x) - F(x^*)) + \sigma^2$$

6. *The sequence of step-size is equal to $\alpha_k = \alpha k^{-\beta}$ with $\beta \in (1/2, 1]$.*

7. *$H = \nabla^2 F(x^*)$ is positive definite and $x \mapsto \nabla^2 F(x)$ is continuous at $x^*$.*

8. *Denote $w_k = \nabla F(x_{k-1}) - g(x_{k-1}, \xi_k)$ and $\gamma_k = \mathbb{E}(w_{k+1} w_{k+1}^T | \mathcal{F}_k)$. There is a positive definite matrix $\Gamma_k \xrightarrow{k \to \infty} \Gamma$.*

9. *And there exist $\delta, \varepsilon > 0$ such that almost surely $\sup_{k \geq 1} \mathbb{E}(||w_k||^{2+\delta}|\mathcal{F}_{k-1})\mathbf{1}_{||x - x^*|| \geq \varepsilon} < \infty$.*

*Assume that $(H - \kappa I)$ is positive definite where $\kappa = 1_{\beta=1} 1/2\alpha$. Let $(x_k)_{k \geq 0}$ be obtained by the SGD rule, then $\frac{1}{\sqrt{\alpha_k}}(x_k - x^*)$ weakly converge to a multivariate normal $\mathcal{N}(0, \Sigma)$ where $\Sigma$ satisfy the following equation:*

$$(H - \kappa I)\Sigma + \Sigma(H^T - \kappa I) = \Gamma$$

In our settings, we have $x_k = (v\ \lambda)^{(k)}$ and the $\xi_k$ is new data $(x_k, y_k)$. The gradient $g(x_k, \xi_k)$ is $\nabla - \ell(v^{(k)}, \lambda^{(k)}; x^{(k)}, y^{(k)})$. Also, we have set $C_k = I, \alpha = \frac{1}{\zeta}, \beta = 1$ where $\zeta I$ is the lower bound of the Hessian matrix $\mathbf{H}$ at $(v^*, \lambda^*)$. Here, we can see that $\mathbf{H} - \frac{1}{2\alpha}I$ is positive definite, since $\mathbf{H} \succeq \frac{1}{\alpha}I \succ \frac{1}{2\alpha}I$. In the section below, we prove that the nine assumptions hold in our settings. Also, in our settings, $\mathbf{H}$ is symmetric, so $\mathbf{H}^T = \mathbf{H}$. We will omit the transpose of $H$ in the future content.

### B.8.1 Assumptions 1-7

- Assumption 1 holds for our algorithm: each time the sampled $u_i$ is an unbiased estimator of $\nabla \bar{\ell}(v^{(t-1)}, \lambda^{(t-1)})$.

- Assumption 2 holds because of our chosen parameter. The step size is $\eta_t = 1/\zeta \cdot t$ which also satisfies assumption 6.

- Assumption 3 is proved in the subsection B.5.

- Assumptions 5 and 7 are implied by Theorem 4.1 for the bounded step variance and strong convexity.

- From the strong convexity of the Hessian Matrix, we derive that the solution of $\nabla \bar{\ell}$ is unique since $\bar{\ell}$ has an unique minimum. Also, since the Hessian matrix is strongly convex everywhere, it can be deduced that when $\|(v, \lambda)\| \to \infty$, the $\bar{\ell}$ goes to infinity. This proves assumption 4.

### B.8.2 Assumptions 8 & 9

First, we propose another proposition, to state that $(v, \lambda)$ converge almost surely to $(v^*, \lambda^*)$:

**Theorem B.10.** *Proposition 1 in Leluc & Portier (2020): Assumptions 1 to 5 are fulfilled. Then the sequence of iterates $(x_k)_{k \geq 0}$ obtained by the SGD rule in Definition B.8 converges almost surely towards the minimizer $x_k \to x^*$*

Since $(v, \lambda)$ converge almost surely to $(v^*, \lambda^*)$, the $\bar{\ell}$ is twice differentiable, and $x^{(i)}, y^{(i)}$ are i.i.d, then we have that the distribution of $w_k$ converges to the case when $v, \lambda \to v^*, \lambda^*$. So, $\mathbb{E}[w_k w_k^T]$

converges to some matrix $\Gamma$. The matrix $\Gamma$ is positive definite, since it is a positive combination of some matrices of form $vv^T$. Since $n \to \infty$ the parameter $(v, \lambda)$ converge to $(v^*, \lambda^*)$ almost surely, and the gradient of $\bar{\ell}$ at the true parameter is zero, we have

$$\Gamma = \mathop{\mathbb{E}}_{x \sim \mathcal{P}} \mathop{\mathbb{E}}_{y} \left[ (\frac{\partial l(x, y; v, \lambda)}{\partial (v, \lambda)} - \nabla \bar{\ell})(\frac{\partial l(x, y; v, \lambda)}{\partial (v, \lambda)} - \nabla \bar{\ell})^T \right]$$

$$= \mathop{\mathbb{E}}_{x \sim \mathcal{P}} \mathop{\mathbb{E}}_{y} \left[ \left( \frac{\partial l(x, y; v, \lambda)}{\partial (v, \lambda)} \right) \left( \frac{\partial l(x, y; v, \lambda)}{\partial (v, \lambda)} \right)^T \right]$$

when $(v, \lambda) = (v^*, \lambda^*)$. Since the assumption 2.4 gives $\mathbb{E}_{x \sim \mathcal{P}} xx^T \succeq bI$, the rank of $x^{(i)}$ is full. Therefore, the gradients will not all be on the same linear subspace (since the gradient is some scalar times $x^{(i)}$ and $y^{(i)}$, which itself is generated with noise.) Hence, $\Gamma$ is positive definite.

For assumption 9, notice that if we bound $v, \lambda$ by the projection set, the $w, \sigma$ are both bounded. Thus, $\forall (v, \lambda) \in D_r$, the gradient is bounded if we set $\varepsilon$ such that $||(v, \lambda) - (v^*, \lambda^*)||$ to be inside the projection set. Also $\mathcal{P}$ is bounded. So, if $y \sim F_x = \mathcal{N}(w^{*T} x, \sigma^{*2}, S)$, we have

$$\frac{\partial \ell}{\partial v} = (\mathbb{E}_{z \sim Q_x}[z] - y)x, \quad \frac{\partial \ell}{\partial \lambda} = \frac{1}{2}(y^2 - \mathbb{E}_{z \sim Q_x}[z^2])$$

Where $Q_x = \mathcal{N}(w^T x, \sigma^2, S)$. Next, we show that $\mathbb{E}_y \left[ \left\| \frac{\partial \ell}{\partial v} \right\|^4 + \left( \frac{\partial \ell}{\partial \lambda} \right)^4 \right]$ is bounded. We know that the expression can be written as

$$\mathbb{E}_y \left[ \left\| \frac{\partial \ell}{\partial v} \right\|^4 + \left( \frac{\partial \ell}{\partial \lambda} \right)^4 \right] = \frac{\mathbb{E}_{y \sim \mathcal{N}(w^{*T} x, \sigma^{*2})} \left[ \mathbf{1}_{y \in S} \left( \left\| \frac{\partial \ell}{\partial v} \right\|^4 + \left( \frac{\partial \ell}{\partial \lambda} \right)^4 \right) \right]}{\alpha(w^*, \sigma^*, x)^4}$$

$$\leq \frac{1}{a^4} \mathbb{E}_{y \sim \mathcal{N}(w^{*T} x, \sigma^{*2})} \left[ \left\| \frac{\partial \ell}{\partial v} \right\|^4 + \left( \frac{\partial \ell}{\partial \lambda} \right)^4 \right]$$

Because $\mathbb{E}[z]$ and $\mathbb{E}[z^2]$ are bounded, and $x$ is also bounded. Also, $w^{*T} x$ and $\sigma^*$ are bounded. the expression inside the expectation can be written as a polynomial of $y$ of degree $\leq 8$. Notice that for the polynomial $P(t)$, if $t$ is a given normal distribution $\mathcal{N}(\mu, \sigma)$ where $\mu$ and $\sigma$ are bounded, $\mathbb{E}_y(P(t))$ is bounded. Thus, we have $\mathbb{E}_y \left[ \left\| \frac{\partial \ell}{\partial v} \right\|^4 + \left( \frac{\partial \ell}{\partial \lambda} \right)^4 \right]$ is bounded if we choose $x$ in the projection set.

Therefore, $||w_k||^4 = ||\frac{\partial \ell(v, \lambda, x, y)}{\partial (v, \lambda)} - \nabla \bar{\ell}||^4 \leq 8(||\frac{\partial \ell(v, \lambda, x, y)}{\partial (v, \lambda)}||^4 + ||\nabla \bar{\ell}||^4)$ is also bounded, since the bound of the latter term can be derived by the bounded step variance in 4.1.

Therefore, when we set $\varepsilon$ such that $||(v\ \lambda) - (v^*\ \lambda^*)||$ to be inside the projection set, and $\delta = 2$, we can maintain a bounded expectation.

### B.8.3   Final Estimation

Since all assumptions hold, we know that the difference of the estimator and true parameter $\left( \sqrt{k^{-1}/\zeta} \right)^{-1} ((v\ \lambda) - (v^*\ \lambda^*))$ is asymptotically normal, and that the variance is given by equation

$$(\mathbf{H} - \kappa I)\Sigma + \Sigma(\mathbf{H}^T - \kappa I) = \Gamma$$

Where $\mathbf{H}$ is the Hessian matrix at $(v^*, \lambda^*)$.

Notice that the $(v, \lambda)$ converge to $(v^*, \lambda^*)$ almost surely, so we can apply the plug-in confidence interval. We can substitute the true estimates by the empirical estimates as justified by the Slutsky's theorem. Notice that the parameter $\hat{\mathbf{H}} = \mathbf{H}(\hat{v}, \hat{\lambda})$ converges to the true parameter $\mathbf{H}$ (because $v, \lambda$ is converging by Theorem B.10) And $\hat{\Gamma}$ (defined below) converges to $\Gamma$ (by the last subsection.) Therefore, we can apply Slutsky's Theorem and we can substitute the estimated parameters for true parameters to calculate the confidence region. We can substitute $\mathbf{H}$ to $\hat{\mathbf{H}}$ where $\hat{\mathbf{H}} = \mathbf{H}(\hat{v}, \hat{\lambda})$ is the empirical Hessian matrix evaluated at $(\hat{v}, \hat{\lambda})$. Also, the empirical $\Gamma$ can be evaluated by $\Gamma(\hat{v}, \hat{\lambda})$, where

$$\Gamma(v, \lambda) = \frac{1}{n} \sum_{i=1}^{n} \frac{\partial l(v, \lambda; x^{(i)}, y^{(i)})}{\partial (v, \lambda)} \frac{\partial l(v, \lambda; x^{(i)}, y^{(i)})}{\partial (v, \lambda)}^T$$

That is, we can write
$$\sqrt{n}(\hat{\theta} - \theta') \to \mathcal{N}(0, \zeta^{-1}\Sigma)$$
Where $(\hat{\mathbf{H}} - \kappa I)\Sigma + \Sigma(\hat{\mathbf{H}} - \kappa I) = \Gamma(\hat{v}, \hat{\lambda})$, and $\theta$ is the joined vector of $v, \lambda$. Notice that here we are converging to a random matrix, so we interpret this estimation as
$$\zeta^{1/2}\Sigma^{-1/2}\sqrt{n}(\hat{\theta} - \theta') \to \mathcal{N}(0, I)$$
By the delta method, since we have $(w, \sigma) = (v/\lambda, 1/\lambda)$, we know that
$$\nabla(\theta) = J^T = \left. \frac{\partial(w, \sigma^2)}{\partial(v, \lambda)} \right|_{v=\hat{v}, \lambda=\hat{\lambda}} = \begin{pmatrix} 1/\hat{\lambda}I & 0 \\ -\hat{v}^T/\hat{\lambda}^2 & -1/\hat{\lambda}^2 \end{pmatrix} = \begin{pmatrix} \hat{\sigma}^2 I & 0 \\ -\hat{w}^T\hat{\sigma}^2 & -\hat{\sigma}^4 \end{pmatrix}$$

So we have (now $w, \sigma^2$ are written in terms of $\theta$)
$$\sqrt{n}(\hat{\theta} - \theta) \sim \mathcal{N}(0, J(\zeta^{-1}\Sigma)J^T)$$
Or, in other terms, let $S = J(\zeta^{-1}\Sigma)J^T$, we have
$$\sqrt{n}S^{-1/2}(\hat{\theta} - \theta) \sim \mathcal{N}(0, I)$$
Thus, Theorem 5.1 is proved. In addition, this yields
$$(\hat{\theta} - \theta)^T (\frac{1}{n}J(\zeta^{-1}\Sigma)J^T)^{-1}(\hat{\theta} - \theta) \sim \chi_{k+1}$$

Let $q_\alpha$ is the $1 - \alpha$ quantile of the distribution $\chi_{k+1}$, so we have the confidence region
$$(\hat{\theta} - \theta)^T (J(\zeta^{-1}\Sigma)^{-1}J^T)^{-1}(\hat{\theta} - \theta) \leq q_\alpha/n$$

Where we can calculate
$$J^{-1} = \begin{pmatrix} 1/\hat{\sigma}^2 I & -\hat{w}/\hat{\sigma}^4 \\ 0 & -1/\hat{\sigma}^4 \end{pmatrix}$$

To make it clearer, we define $R = J^{-1}$. So Or, write back to $\hat{w}, \hat{\sigma}$, as
$$\left( \begin{pmatrix} w \\ \sigma^2 \end{pmatrix} - \begin{pmatrix} \hat{w} \\ \hat{\sigma}^2 \end{pmatrix} \right)^T R^T \Sigma^{-1} R \left( \begin{pmatrix} w \\ \sigma^2 \end{pmatrix} - \begin{pmatrix} \hat{w} \\ \hat{\sigma}^2 \end{pmatrix} \right) \leq \frac{q_\alpha}{\zeta n}$$

## C   Experiment Setup

In our main paper, we provide theoretical guarantees for truncated linear regression with unknown noise variance and test our procedure on synthetic data. Here, we give an overview of how we set up these experiments. In our experimental section, we mention that we perform each experiment for an specified number of trials. For each algorithm a trial means something different. For the Croissant & Zeileis (2018) experiments, we write the dataset of interest to a csv file, which is then read in an R script and the procedure is run. For Daskalakis et al. (2019) and our algorithm, a trial is considered complete in a of couple ways. First off, every 100 steps, we check the $L^2$ norm of the validation set's gradient. If its magnitude is less than $1e - 1$, we terminate the procedure, and return the current estimates. However, if after taking 2500 gradient steps, the validation set's gradient $L^2$ norm is greater than $1e - 1$, we re-run the stochastic process. We do this a maximum of three times, and return the trial with the smallest gradient.

For these experiments, we use a PyTorch SGD optimizer, starting our procedure with a learning rate of $1e - 1$, and decaying it at a rate of .9 every 100 gradient steps.

All of the experiments performed in this paper were performed on a 15-inch MacBookPro with a 2.2 GHz 6-Core Intel Core i7 with 16 GB of memory. It is of note to mention that 2500 gradients steps with batch size 10 ran in a maximum of 3 seconds for the experiments.

# D Code

We provide code for our experiment at the following GitHub repository: https://github.com/pstefanou12/Truncated-Regression-With-Unknown-Noise-Variance-NeurIPS-2021.

Below, we provide the gradient that we used for conducting all of our experiments.

```python
class TruncatedUnknownVarianceMSE(ch.autograd.Function):
    """
    Computes the gradient of negative population log likelihood for
    truncated linear regression with unknown noise variance.
    """
    @staticmethod
    def forward(ctx, pred, targ, lambda_, phi):
        ctx.save_for_backward(pred, targ, lambda_)
        ctx.phi = phi
        return 0.5 * (pred.float() - targ.float()).pow(2).mean(0)

    @staticmethod
    def backward(ctx, grad_output):
        pred, targ, lambda_ = ctx.saved_tensors
        # calculate std deviation of noise distribution estimate
        sigma = ch.sqrt(lambda_.inverse())
        stacked = pred[None, ...].repeat(args.num_samples, 1, 1)
        # add noise to regression predictions
        noised = stacked + sigma * ch.randn(stacked.size())
        # filter out copies that fall outside of truncation set
        filtered = ctx.phi(noised)
        z = noised * filtered
        lambda_grad = .5 * (targ.pow(2) - (z.pow(2).sum(dim=0) /
        (filtered.sum(dim=0) + args.eps)))
        """
        multiply the v gradient by lambda, because autograd computes
        v_grad*x*variance, thus need v_grad*(1/variance) to cancel
        variance factor
        """
        out = z.sum(dim=0) / (filtered.sum(dim=0) + args.eps)
        return lambda_ * (out - targ) / pred.size(0),
        targ / pred.size(0), lambda_grad / pred.size(0), None
```

Listing 1: Truncated version of mean squared-error loss

# E Future Work

There are many ways to build upon our work. One thing to point out is that, we use an SGD framework, while Croissant & Zeileis (2018) uses an analytic gradient and Hessian provided by the Henningsen & Toomet (2011) package and the Newton-Raphson method. Since we do not explicitly calculate integrals in our method, our framework could be used to train non-linear models, including neural networks. Further, now that there are two methods to learn from truncated linear models, it would be interesting to explore an actual example where a dataset has been *truncated* or *censored* due to uncontrollable circumstances. With multiple methods for learning from truncated samples, it would interesting to see what results each of the method's give. An interesting field to explore would be environmental sciences, as there are a lot of examples where there is truncation due to measurement instrumentation failure or natural causes. One last to ponder would be what theoretical guarantees can be derived for an algorithm in the *censored* setting. We emphasize that our algorithm for truncated data will work in the censored setting, but with additional knowledge of all the covariate features, can we design an algorithm with better error and/or run-time bounds?