# OpenReview forum: "Efficient Truncated Linear Regression with Unknown Noise Variance"
_NeurIPS.cc/2021/Conference — NeurIPS 2021 Poster_

### Official Review · Reviewer_gvjA · 2021-07-09

**Rating:** 7
**Confidence:** 3

**Summary:**

In this paper, the authors investigate truncated linear regression, i.e., inferring the linear model parameters when only a subset of the data can be observed. The primary contribution is that they provide error bounds on the estimators for model parameters when the noise variance is unknown and show that there exists a polynomial time algorithm to find the estimate (noise variance included).   They validate their results on synthetic data.

**Main Review:**

1. Originality/significance: the topic of truncated regression is an interesting one that comes up in practice (whether or not practitioners consider it) and is certainly worthy of study. As the authors acknowledge and adequately cite, their work is an extension of Daskalakis et al. Although the main ideas are drawn from the work of others, the novelty of having unknown variance is significant and interesting.

2. Quality: in general, the quality of the work is good with technical rigor and writing that is easy to understand (with a caveat, see clarity below). The quality of the synthetic experiments leaves something to be desired since it isn't clear if performance is based on well chosen examples or the merit of the method. More details can be found in comments below. Since there is a theoretical basis to their work, this might not be much of a concern.

3. Clarity: although the writing is clear, the arguments and structure of the paper are circuitous and at times hard to follow. For instance, in section _3 Log likelihood functions and feasibility_ on line 167 it states that the log likelihood function is defined in Section 4.1; why include a section title as such only to defer until later? There are other issues such as restating assumptions in a similar form multiple times. The layout/placement of theorems and their connection to one another is difficult to follow. For instance, include theorems 3.5 and 3.6 in Section 4 and move computation considerations to discussion of algorithms. The paper's structure seems to be its primary weakness at the moment.

### Minor comments
- line 24: range instead of rage?
- line 53: "needed" rather than "need"
- line 203: what is "folklore boosting"? Is it just what's described in the next sentence?
- line 255: where exactly in the appendix can we find details to solve the computational problem?
- line 264: is "initial point" supposed to be "optimal point" here?

### Major comments, clarifying questions
- assumption 1.1: is $\beta$ generally known _a priori_ or is it chosen to be conservative? Given that theorem 1.4 depends exponentially on it, this seems like it could be an issue
- line 93: is the strong convexity bounded at the optimum alone or in some neighborhood about it?
- Section 2: much of the detail in this section is repetitive and could be treated more thoroughly/merged with the introduction. The detail is not so technical that it requires a new section.
- Section 3: the section title is _Log likelihood functions and feasibility_ but the log likelihood isn't introduced until section 4.1. Discussion of the algorithm here also seems out of place since the form of the objective, its gradient, and constraints haven't been introduced yet. Maybe introduce the log likelihood and feasible set here wait on discussion of algorithms until after the form of the problem has been set.
- Section 4.1: see comment above. Also, in line 240, explicity stating the pdf for the truncated Gaussian is helpful for general reader and will also clarify where the MLE comes from.
- Section 4.2: specify where exactly in the supplement computation problems are discussed. Perhaps spend more time discussing computational issues.
- Section 6:
    * Experiment for figure 1: since $w*$ is fixed, what is it here and does the performance vary considerable for different $w*$? Also, it seems that given the symmetry in the problem, less might be lost by truncation. Does the performance deteriorate if truncated on both tails, e.g., $S = \{-1.5 \le y \le  2\}$? (is "unknown" is fig 3-5 in supplement the proposed algorithm )
    * I think figure 2a is intended to address issue above, but it is impossible to draw any conclusions from the image. Please include an image or data illustrating that is more easily interpreted.
    * Experiments for figure 2: using different noise model, i.e., Laplacian, changes the MLE (example no longer agrees with theory) so I don't necessarily expect the algorithm to perform well. However, it is surprising that _truncreg_ outperforms as it does. What is the reason for this?
    * Reasonably, what size problem can be solved with this method? What is maximum dimension attempted

=================================
UPDATE

Thank you to the authors for their response to reviewer comments; they have helped improve understanding. I also agree that "statistical efficiency" is confusing and should be changed.

With the changes mentioned above, I have improved my rating from 6 to a 7.



**Time Spent Reviewing:**

10

---

> ### Author Response · Authors · 2021-08-10
> **Rebuttal**
>
> ***Minor Comments:***
> Thank you for catching the typo in the title of Section 3 and for the rest of the typos! We will definitely incorporate these suggestions in improving the organization of the paper! We commit to fix all these for the final version.
>
> ***assumption 1.1 ... an issue:***
> In many cases, this parameter can be estimated by estimating the portion of the samples that are expected to be truncated from the truncation mechanism. If such information is not available then some hyperparameter tuning techniques might be possible to be applied in order to avoid a very conservative estimate. We will discuss this issue in more detail after Assumption 1.1 in the final version.
>
> ***line 93: ... about it?:***
> We used the strong convexity inside the set D_r for inequality (3.2) (which utilized theorem 3.6), and used the strong convexity at the optimum for inequality (3.1) (which utilized theorem 3.5)
>
> ***Section 2, 3, 4, 4.1, 4.2:***
> Thank you for feedback on our paper's clarity! We commit to resolving these clarity issues in the paper's next iteration.
>
> ***Section 6:***
> For Figure 1, we randomly selected w* by sampling from a Uniform(-1, 1) for each weight coefficient and then the bias term. For the first experiment, we explored how varying the model’s noise variance affects our procedure’s performance. We do not use interval truncation for the first experiment because the ***truncreg*** package only supports left *or* right truncation. Thus, we cannot compare the two packages for the proposed truncation set; -1.5<=y<=2.
>
> Thank you for your feedback on Figure 2a, we will be sure to either provide a more thorough experiment or remove it entirely from the next iteration of the paper.
>
> Yes, “unknown” in the supplementary material is our proposed algorithm. We are sorry about the discrepancy, and will be sure to correct this in the next iteration of our paper. Something to note about the Laplacian experiments was that our methods did reduce a significant amount of bias in the bias and weight coefficients. However, the variance results were not particularly good. We believe that this makes sense due to the fat-tailedness of the Laplacian distribution.
>
> We are unsure why the ***truncreg*** package is extremely stable and produces very consistent results.
>
> We have run an experiment in 50 dimensions, where the initial weight coefficients are a 50 dimension vector of ones. We then applied the following truncation set S = [(-20, -5), (-3, 0), (8, 10)]. We then varied the number of samples that fall within the truncation set over the range [10, 5000]. The resulting figure resembles Figure 2c. The results from our experiment show that our algorithm is stable in 50 dimensions.
>
> I checked the code that was submitted and once you have the dependencies installed, it works as described in the paper. We submitted the repo url in the supplementary materials. If you are interested at all in looking at the code and maybe increasing dimensions, etc., and have any questions, please let me know. The code should run efficiently on your local device.

---

### Official Review · Reviewer_Hgq6 · 2021-07-16

**Rating:** 6
**Confidence:** 4

**Summary:**

The paper extends the study of Daskalakis et al. (2019) into the setting where the noise variance is unknown, and gives two theoretical contributions. First, the paper proves the convergence of the projected SGD algorithm when applied to minimizing the population negative log-likelihood function. Second, the paper provides the asymptotic normality of their estimator, from which the confidence regions for the estimator can be computed.



**Limitations And Societal Impact:**

I have not known any potential negative societal impact of the present work.

**Main Review:**

As a direct extension of Daskalakis et al. (2019), the novelty of the paper seems moderate, while this is somewhat compensated by a clear discussion on the difference from Daskalakis et al. (2019) in the beautifully written intro, and by nontrivial theoretical development using statistics and optimization.

Even though the authors claimed their efforts in proofreading the paper, there are several issues that concern me in writing:
  1. The authors use both n and N to represent the number of samples. For example, it is n in Assumptions 1.3, but it is N at Lines 126 and 139. Moreover, they use N to represent normal distribution at Lines 335, 342, and 343. Finally, they use N to mean the number of iterations at Line 68 (I guess), and n at Line 346 to mean convergence rate. In Algorithm 1, do the authors mean n samples and N iterations? These are very confusing, please clarify.
  2. There are certain repetitions on writing. For example, the assumptions are repeated. In Theorems 3.5 and 3.6, (i) and (iii) are repeated. In Theorem 3.6, one can simply reference (i) and (iii) of Theorem 3.5. Lines 171-173 can be simply replaced by Lines 247-256.
  3. In assumption 2.4, there is a full stop missing. And there is a typo at Line 243.
  4. Section 3 is called "Log likelihood functions and feasibility". But the content is about projected SGD and Theorem 3.2.
  5. I would define the log-Likelihood function much earlier, rather than in Section 4.1, and I would put the computation of its gradient and Hessian into the supplementary.


I have the following two questions.
  1. Please explain the definition of D_r. In particular, how does the semidefinite constraint arise? What is the intuition here or does it follow directly from Daskalakis et al. (2019)?
  2. At the beginning of Section 4.5, please explain why to split the samples into 3 parts. Is it more natural to compute the OLS and run the SGD using all samples? If this can not be explained theoretically, please have an ablation study on splitting or not.

------ UPDATE ------

The authors have addressed my concerns. Thus I raised my score to 6.


**Time Spent Reviewing:**

8

---

> ### Author Response · Authors · 2021-08-10
> **Author Response for Reviewer Hgq6**
>
> **Even though the authors claimed their efforts in proofreading the paper, there are several issues that concern me in writing:
> The authors use both n and N to represent the number of samples. For example, it is n in Assumptions 1.3, but it is N at Lines 126 and 139. Moreover, they use N to represent normal distribution at Lines 335, 342, and 343. Finally, they use N to mean the number of iterations at Line 68 (I guess), and n at Line 346 to mean convergence rate. In Algorithm 1, do the authors mean n samples and N iterations? These are very confusing, please clarify.**
>
> These are typos, thanks for catching them, we mean $n$ everywhere.
>
> **There are certain repetitions on writing. For example, the assumptions are repeated. In Theorems 3.5 and 3.6, (i) and (iii) are repeated. In Theorem 3.6, one can simply reference (i) and (iii) of Theorem 3.5. Lines 171-173 can be simply replaced by Lines 247-256.**
>
> Thank you for this observation! We will definitely remove these superfluous repetitions for the next iteration of our paper. With the additional space, we will include the semi-synthetic experiments from our supplementary material into the main version of our paper.
>
> **In assumption 2.4, there is a full stop missing. And there is a typo at Line 243.**
>
> Thank you for pointing this out. We will fix this.
>
> **Section 3 is called "Log likelihood functions and feasibility". But the content is about projected SGD and Theorem 3.2.**
>
> Thank you for your feedback. We will change the title to: Algorithm and Main Results.
>
> **I would define the log-Likelihood function much earlier, rather than in Section 4.1, and I would put the computation of its gradient and Hessian into the supplementary.**
>
> Thank you for your feedback, we will include this change in the next iteration of our paper.
>
> **I have the following two questions.
> Please explain the definition of D_r. In particular, how does the semidefinite constraint arise? What is the intuition here or does it follow directly from Daskalakis et al. (2019)?**
>
> The semidefinite constraint part is indeed derived from Daskalakis et al. (2019). Roughly the idea of these constraints is that the error of the estimation in any direction depends on the variance of the matrix $\sum x_i x_i^T$ in this direction. To express this difference across different directions we need exactly these semidefinite constraints. We will add a paragraph to give some intuition about the $D_r$ set after its definition.
>
> **At the beginning of Section 4.5, please explain why to split the samples into 3 parts. Is it more natural to compute the OLS and run the SGD using all samples? If this can not be explained theoretically, please have an ablation study on splitting or not.**
>
> Indeed in practice it would be better to use all the samples both for OLS and for SGD but theoretically this introduces dependencies. If the initialization depends on the samples that are later used to run the SGD then these dependencies are difficult to analyze. For this reason we follow the simplest and safest strategy of assuming fresh samples for each one of the tasks and analyze each task independently without having to worry about the dependencies from using the same samples. We will make sure to clarify this even further in the beginning of Section 4.5 and highlight the difference in the experimental section.

---

### Official Review · Reviewer_nLxV · 2021-07-29

**Rating:** 7
**Confidence:** 3

**Summary:**

# Summary

The paper studies truncated linear regression when the additive error follows a zero-mean normal distribution with unknown variance.
This setting generalizes the work of Daskalakis-Gouleakis-Tzamos-Zampetakis-2019, who assumed the knowledge of the variance of error.
This work provides two statistical rates for the joint estimation of the regressor (w) and the noise variance (\sigma) in Theorem 3.2.
The rates in Theorem 3.2 are achieved using a SGD-without-replacement algorithm on the negative log likelihood after a suitable re-parameterization, which is also used in Daskalakis-Gouleakis-Tzamos-Zampetakis-2019.



**Limitations And Societal Impact:**

Yes

**Main Review:**

I think the paper studies an important problem of generalizing the work of Daskalakis et al. (2019) to the setting of unknown variances. As the paper claims, this requires non-trivial ideas and natural approaches fail.
As the joint optimization of w and sigma is non-convex, the paper performs a re-parameterization of the variables to induce convexity. This re-parameterization is similar to the one used in Daskalakis et al. (2018).

Overall, I think the paper has a few good ideas but needs additional work in terms of presentation and clarity. Some important details are not clear and  details are hard to follow. Thus I recommend weak reject for now, and I will improve my score if the following comments are addressed:


+ Lines 104-131: I really liked the idea of  explaining why natural approaches to extend Daskalakis et al. (2019) fail. However, a lot of details in these lines are not clear to me. In particular, the sentences: (i) "It is clear that (Line 118)..." and (ii) "It is easy to see that (Line 125) ...". I would appreciate if the authors can explain these cases as formal claims in the supplementary text with the omitted details.

+ Theorem 1.5 (Asymptotic normality): What does it mean for a random variable $X$ to converge to a Gaussian with a covariance that depends on $X$? That is, $X \to^d \mathcal{N} (\mu, \Sigma_X^2)$? How can the covariance matrix itself be random? I skimmed through Leluc and Portier (2020) and they show convergence to a normal distribution that depends on the true optimizer.


**Minor points**

+ As Leluc and Portier (2020) heavily influences the  asymptotic normality result, they should be cited appropriately in the main paper.

+ The large length of some sentences affect readability; for example, Line 17-21. Line 28-33.

+ Add a citation for the Line 157-158.

+ Please add the details for the convexity of $D_r$ after Definition 3.1 (or point to the supplementary material). It is not obvious (to me at least) at the first look.

+ Refer to Algorithm 3.1 in the beginning of Theorem 3.5. It is not clear in the beginning what is $u^{(t)}$.


+ I am not sure if the phrase "statistically efficient" (without any quantifier) can be used to describe the rates in Theorem 3.2 as they may not be optimal, i.e., may not achieve efficiency.

+ Add a small claim for the typical values of b (and the corresponding sample complexity) in Assumption 2.4 for the (rescaled) multivariate normal distribution.

+ Line 177: What is the difference in the notation?



+ Assumption 1.2 seems stronger than Assumption 1 in Daskalakis et al. (2019).

+ Lines 101-103: How does Lemma 7 in Daskalakis et al. (2019) imply the claim in Lines 101-103?



+ Line 35: Typo: needed,

+ Define $H_x$ in (4.3)?

+ Line 253-254: Where is this in supplementary material. In general, please point to the specific section when referring to the supplementary material.

+ Line 264: 'initial' should be replaced by 'optimal'.

# Update

I thank the authors for their detailed response and discussion.I am thus raising my score from 5 to 7. Repeating the sentiment from my original review, I would strongly suggest improving the presentation and clarity ---both technical and non-technical aspects. If the paper is accepted or for the new submission, I hope the authors would follow the suggestions pointed out in all of the reviews to improve clarity and presentation. In particular, I suggest the following:
1. Add as formal claims why obvious approaches won't work (with omitted details in the supplementary material).
2. Improve the notation for asymptotic normality
3. Changing the phrase statistical efficiency
4. Improve the structure of the paper as other reviewer suggested
5. Citing Leluc and Portier before stating Theorem 5.1 and Theorem 1.5.
6. And other minor/major comments

**Time Spent Reviewing:**

5

---

> ### Author Response · Authors · 2021-08-10
> **Author Response for Reviewer nLxV**
>
> **Lines 104-131: I really liked the idea of explaining why natural approaches to extend Daskalakis et al. (2019) fail. However, a lot of details in these lines are not clear to me. In particular, the sentences: (i) "It is clear that (Line 118)..." and (ii) "It is easy to see that (Line 125) ...". I would appreciate if the authors can explain these cases as formal claims in the supplementary text with the omitted details.**
>
> In line 118, if $\tau$ is larger, intuitively the huge mass of the distribution is more concentrated in a small region around tau, as the pdf of the truncated normal decreases sharply. We can intuitively regard the distribution “closer” to the point mass of $\tau$, and the variance is smaller. In fact, the variance is smaller than $2/\tau^2$. We next give the exact calculation to verify this result and we commit to add them in the supplementary material of our paper.
>
> The variance of that distribution $\mathcal{N}(0,1,\{x>\tau\})$ is smaller or equal to the expectation of $(x-\tau)^2$, which we can calculate it by: (Let $q_{\tau}=\int_{\tau}^\infty e^{-x^2/2}\mathrm{d}x$)
>
> $$ \mathbb{E}_{x\sim \mathcal{N}(0,1,\{x>\tau\})}[x^2]$$
>
> $$=\int_\tau^\infty \frac{e^{-x^2/2}}{q_{\tau}}{(x-\tau)^2}\mathrm{d}x=\int_\tau^\infty \frac{e^{-x^2/2}}{q_{\tau}}\int_{\tau}^x{2(s-\tau)}\mathrm{d}s\mathrm{d}x $$
>
> $$= \int_\tau^\infty {2(s-\tau)}\int_{s}^\infty \frac{e^{-x^2/2}}{q_{\tau}}\mathrm{d}x\mathrm{d}s = \int_\tau^\infty {2(s-\tau)}\mathbb{P}(x>s|x>\tau,x\sim\mathcal{N}(0,1))\mathrm{d}s$$
>
> We claim that $\mathbb{P}(x>s|x>\tau,x\sim\mathcal{N}(0,1))<e^{-\tau (s-\tau)}$ Notice that $x^2\le (\tau+s)x-\tau s$ if $x\in [\tau,s]$ and $x^2\ge (\tau+s)x-\tau s$ otherwise. Let $A = \int_{\tau}^s e^{-x^2/2}\mathrm{d}x$ and $B = \int_s^\infty e^{-x^2/2}\mathrm{d}x = q_\tau -A$. So we have $\mathbb{P}(x>s|x>\tau,x\sim\mathcal{N}(0,1))=\frac{B}{A+B}=\frac{1}{A/B + 1}$ Thus, we have
> $$A=\int_{\tau}^s e^{-x^2/2}\mathrm{d}x\ge \int_{\tau}^s e^{-((\tau+s)x-\tau s)/2}\mathrm{d}x:=A', B=\int_s^\infty e^{-x^2/2}\mathrm{d}x\ge \int_{\tau}^s e^{-((\tau+s)x-\tau s)/2}\mathrm{d}x:=B'$$
> So we have $A/B\ge A'/B'$, Therefore, we get
>     $$\mathbb{P}(x>s|x>\tau,x\sim\mathcal{N}(0,1))=\frac{1}{A/B + 1}\le =\frac{1}{A'/B' + 1}= \frac{B'}{A' + B'}= \frac{\int_s^\infty e^{-((\tau+s)x-\tau s)/2}\mathrm{d}x}{\int_{\tau}^\infty e^{-((\tau+s)x-\tau s)/2}\mathrm{d}x}$$
>     $$=\frac{e^{\tau s/2}\frac{2}{\tau+s}e^{-s((\tau+s)x)/2}}{e^{\tau s/2}\frac{2}{\tau+s}e^{-\tau((\tau+s)x)/2}}=e^{-\frac{\tau+s}{2}(\tau-s)}\le e^{-\tau (s-\tau)}$$
> Therefore
> $$\mathbb{E}_{x\sim \mathcal{N}(0,1,\{x>\tau\})}[x^2]\le \int_\tau^\infty {2(s-\tau)}\mathbb{P}(x>s|x>\tau,x\sim\mathcal{N}(0,1))\mathrm{d}s\le\int_\tau^\infty {2(s-\tau)}e^{-\tau (s-\tau)}\mathrm{d}s =\frac{2}{\tau^2}$$
>
> The claim in line 125 can be derived as follows: In both algorithms, with feeding a sufficient number of samples, will return hat(w) and hat(w)’ close to zero. Intuitively, (2) (line 124) is going to find a result of small variance, and the algorithm Daskalakis et. al. will behave like OLS. Therefore, we may get a result of $\hat(w_0)’$ close to its mean. This makes the sum of the remaining square be at the magnitude of the variance, and the right hand side is O(1/\tau^2). (1) (line 122) the algorithm in Daskalakis et. al. (2019) will behave as in the new algorithm in this paper (since this algorithm optimizes both weight and variance, and if we are given correct variance, it will return a value of true weight.) Therefore the algorithm will return hat(w_0) close to 0, and the left hand side is close to $|1-\mathbb{E}(y_i^2)|$, approximately tau^2-1, which is much greater than the right hand side.
>
>
> **Theorem 1.5 (Asymptotic normality): What does it mean for a random variable to converge to a Gaussian with a covariance that depends on ? That is, how can the covariance matrix itself be random? I skimmed through Leluc and Portier (2020) and they show convergence to a normal distribution that depends on the true optimizer.**
>
> It is indeed easier to write the covariance of the Gaussian in the limit with respect to the true optimizers, but this result is not a useful one when you estimate the confidence intervals in practice. For this reason we use Slutsky's theorem and we replace the true optimizer with the estimated one to be able to provide a confidence region that is actually computable from data in practice. We will add a remark to clarify this point in the final version.
>
> **Minor points
> As Leluc and Portier (2020) heavily influences the asymptotic normality result, they should be cited appropriately in the main paper.**
>
> Thank you for pointing this out. We will be sure to correct this in our paper’s next iteration.
>
> **The large length of some sentences affect readability; for example, Line 17-21. Line 28-33.**
>
> Thank you for your feedback on our readability. We will fix this for the next version.
>
> **Add a citation for the Line 157-158.**
>
> We will add citations, as this setting also appears in Daskalakis et. al. 2019
>
> **Please add the details for the convexity of after Definition 3.1 (or point to the supplementary material). It is not obvious (to me at least) at the first look. Refer to Algorithm 3.1 in the beginning of Theorem 3.5. It is not clear in the beginning what is**
>
> We will add it to supplementary material to prove that this projection set is convex. To see this convexity in short, we can regard the upper bound of the covariance matrix as an infinite intersection of the convex sets (actually, they are ellipsoids), each of the convex set (ellipsoid) is produced by multiplying a unit vector on the left and right of the matrices in that semidefinite inequality. Thus the upper bound of the semidefinite matrix produces a convex set. The variance constraint is a region between two hyperplanes (thus also convex), the convexity of the bound constraint on w can be derived using triangle inequality. Therefore, the constraint D_r is an intersection of three convex sets, thus convex.
>
> **I am not sure if the phrase "statistically efficient" (without any quantifier) can be used to describe the rates in Theorem 3.2 as they may not be optimal, i.e., may not achieve efficiency.**
>
> Indeed we agree that this phrase is confusing. We refer to “efficiency” in the Theoretical Computer Science sense, i.e., the number of samples required is polynomial in $1/\epsilon$. The “statistically” in the above phrase refer to the fact that we are talking about the sample complexity.
>
> **Add a small claim for the typical values of b (and the corresponding sample complexity) in Assumption 2.4 for the (rescaled) multivariate normal distribution.**
>
> We will add a comment about this.
>
> **Line 177: What is the difference in the notation?**
>
> These are the samples dealing with the initialization
>
> **Assumption 1.2 seems stronger than Assumption 1 in Daskalakis et al. (2019).**
>
> Yes it is.
>
> **Lines 101-103: How does Lemma 7 in Daskalakis et al. (2019) imply the claim in Lines 101-103?**
>
> We do not cite Daskalakis et al. (2019) in lines 101-103. Instead we cite Daskalakis et al. (2018), where they solve the problem of estimating the parameters of a truncated multivariate gaussian distribution.
>
> **Line 35: Typo: needed,**
>
> Thanks for pointing this out, We will fix that.
>
> **$H_x$ Define in (4.3)?**
>
> We will clarify this and fix this in the next iteration.
>
> **Line 253-254: Where is this in supplementary material. In general, please point to the specific section when referring to the supplementary material.**
>
> We will fix this.
>
> **Line 264: 'initial' should be replaced by 'optimal'.**
>
> Thanks for pointing out, We will fix this

---

> > ### Comment · Reviewer_nLxV · 2021-08-27
> > **Followup**
> >
> > I thank the authors for their detailed response. I have the following questions:
> >
> > 1. I still do not understand what it means for a random variable $X_n$ to converge to a Gaussian with a covariance that depends on $X_n$? The covariance is supposed to be a constant number. Can the authors please help me understand what the Theorem 5.1 means?
> > (Also I would suggest adding subscript $t$ to $(\hat{w}, \hat{\sigma^2})$ in Theorem 5.1 and be explicit about $t \to \infty$ as the asymptotic regime)
> >
> > 2. As the phrase "statistical efficiency" is confusing, please change its use in the paper.

---

> > > ### Author Response · Authors · 2021-08-27
> > > **Response for follow-ups**
> > >
> > > **1. I still do not understand what it means for a random variable $X_n$ to converge to a Gaussian with a covariance that depends on $X_n$? The covariance is supposed to be a constant number. Can the authors please help me understand what the Theorem 5.1 means? (Also I would suggest adding subscript $t$ to $(\hat{w},\hat{\sigma}^2)$ in Theorem 5.1 and be explicit about $t\to\infty$ as the asymptotic regime)**
> > >
> > > The covariance depends on the true optimizer. However, here we are concerned about the regression’s confidence intervals, which we calculate using only the estimated model parameters. For our proof in Theorem 5.1, we cite Theorem 2 from LeLuc & Portier 2020. In their theorem, the asymptotic covariance matrix is a definite matrix, which calculated by $\Gamma$ (the expectation of the gradient Matrix), $\mathbf{H}$ (the Hessian Matrix), etc. For our confidence interval calculations, we use the same assumptions, but instead we use an estimated value for the approximation for the matrices $\Gamma, \mathbf{H}$ (that is, $\Gamma(v,\lambda), \hat{\mathbf{H}}$), etc. We used this due to the convergence of the matrices and the Slutsky’s theorem. We appreciate your additional comments about our notation.
> > >
> > > We will add the additional $t$ subscript, and note that we are looking at the estimated parameters as $t$ tends to infinity. Thank you for your feedback here, we will clarify this point in the next iteration. Please let us know if you would like us to clarify anything else.
> > >
> > > **2. As the phrase "statistical efficiency" is confusing, please change its use in the paper.**
> > >
> > > Thank you for your  feedback on this phrase, we will change or remove it in the next iteration.

---

> > > > ### Comment · Reviewer_nLxV · 2021-08-28
> > > > **Still not clear**
> > > >
> > > > Thank you for your response.
> > > >
> > > > + But I still don't see how Slutsky's theorem is applicable here, and how it allows you to replace $H$ and $\Gamma$ by $\hat{H}$. The proof in the supplementary material does not contain enough details. In short, Theorem 5.1 does not parse for me.
> > > >
> > > > + I can see how Corollary 5.2 can be true not clear about Theorem 5.1. By the way, what is $\zeta_n$ in Corollary 5.2?

---

> > > > > ### Author Response · Authors · 2021-08-28
> > > > > **Response for additional follow-ups**
> > > > >
> > > > > **But I still don't see how Slutsky's theorem is applicable here, and how it allows you to replace $H$ and $\Gamma$ by $\hat{H}$. The proof in the supplementary material does not contain enough details. In short, Theorem 5.1 does not parse for me.**
> > > > >
> > > > > We have not explicitly cited Slutsky’s theorem. We have two converging matrices, as $n\to\infty$, Slutsky’s theorem allows us to add/subtract/multiply the converging matrices to get more converging matrices. Next, by the continuous mapping theorem (which we will cite and clarify in the next iteration), the solution $\Sigma$ in the equation $(\mathbf{H} - \kappa I)\Sigma + \Sigma(\mathbf{H}^T - \kappa I) = \Gamma$ converges: since the equation mapping $\Gamma$, $H$ to the solution $\Sigma$ is a continuous mapping, and also $H-\kappa I$ is structurally stable (that means it will not touch the boundary of the singularities (thus discontinuities for $\Sigma$)). Hence the solution using the estimated $\Gamma(v,\lambda)$ and $\hat{H}$ converges to the real solution when using true $\Gamma$ and $H$.  We can thus replace $H$ and $\Gamma$ by $\hat{H}$ and $\Gamma(v,\lambda)$, because of the convergence of both $H$ and $\Gamma$, Slutsky’s theorem, and the continuous mapping theorem cited above. The convergence of $H$ and $\Gamma$ is noted in assumptions 7 and 8 in Theorem A.8, and the proof of the assumptions is listed right after that. We will clarify these issues in the next iteration. Please let us know if there is still something that is not clear.
> > > > >
> > > > > **I can see how Corollary 5.2 can be true not clear about Theorem 5.1. By the way, what is $\zeta_n$ in Corollary 5.2?**
> > > > >
> > > > > Sorry for causing the misunderstanding. In the Corollary 5.2 it is $\zeta\times n$, not $\zeta_n$. Please refer further details in A.7.3

---

> > > > > ### Author Response · Authors · 2021-08-29
> > > > > **Additional clarifying for Thm. 5.1**
> > > > >
> > > > > When we say that a random variable $y$ with $y \to \mathcal{N}(0, \Sigma)$, what we mean is that $\Sigma^{-1/2} y \to \mathcal{N}(0, I)$. When the result is written this way then the sequence of random variables only appear in the same side and the limit distribution is fixed. We will update our manuscript accordingly. Please let us know if there is still something around Theorem 5.1 that is not clear.

---

> > > > > > ### Comment · Reviewer_nLxV · 2021-08-29
> > > > > > **This helped!**
> > > > > >
> > > > > > I thank the authors for their detailed response. Writing the convergence in terms of $\Sigma_{n}^{-1/2}X_n \to \mathcal{N}(0,I)$ makes sense to me. I would strongly suggest using the same notation in the main text.

---

### Decision · Program_Chairs · 2021-09-27

**Decision:**

Accept (Poster)

**Comment:**

This paper studies high-dimensional linear regression in a truncated setting, where only examples with labels in a given set are observed.
Learning from truncated samples is a classical topic in statistics that has received renewed interest in the last few years. Prior work had given an efficient algorithm for this problem under the assumption that the additive observation noise has known variance. This paper gives an efficient algorithm for the broader, more realistic, setting that the variance of the noise is *unknown*. Handing unknown variance turns out to require non-trivial new ideas. After extensive discussion internally and with the authors, the reviewers agreed that this contribution merits acceptance to NeurIPS.